# Preleukemic single-cell landscapes reveal mutation-specific mechanisms and gene programs predictive of AML patient outcomes

## Graphical abstract

## Authors

Tomoya Isobe, Iwo Kucinski,
Melania Barile, ..., Brian J.P. Huntly,
Nicola K. Wilson, Berthold Göttgens

## Correspondence

nkw22@cam.ac.uk (N.K.W.),
bg200@cam.ac.uk (B.G.)

## In brief

Isobe et al. profiled 269,048 single-cell transcriptomes of hematopoietic stem and progenitor cells from eight preleukemic mouse models, revealing mutation-specific perturbations in cell abundance, differentiation fate, metabolic activity, and gene expression. They further developed Stem11, a preleukemic lineage perturbation signature correlating with AML patient outcomes.

## Highlights

- Single-cell hematopoietic landscape of eight preleukemic mouse models

- Streamlined pipeline for integrated analysis of single-cell perturbation datasets

- Mutational impact on cell abundance, fate probability, metabolism, and gene expression

- Stem11 lineage perturbation signature predictive of AML patient outcomes

Isobe et al., 2023, Cell Genomics 3, 100426
December 13, 2023 © 2023 The Author(s).

# Cell Genomics

CellPress

## Resource

# Preleukemic single-cell landscapes reveal mutation-specific mechanisms and gene programs predictive of AML patient outcomes

Tomoya Isobe,[1] Iwo Kucinski,[1] Melania Barile,[1] Xiaonan Wang,[1] Rebecca Hannah,[1] Hugo P. Bastos,[1] Shirom Chabra,[1] M.S. Vijayabaskar,[1] Katherine H.M. Sturgess,[1] Matthew J. Williams,[1] George Giotopoulos,[1] Ludovica Marando,[1] Juan Li,[1] Justyna Rak,[1,2] Malgorzata Gozdecka,[1,2] Daniel Prins,[1] Mairi S. Shepherd,[1] Sam Watcham,[1] Anthony R. Green,[1] David G. Kent,[1,3] George S. Vassiliou,[1,2] Brian J.P. Huntly,[1] Nicola K. Wilson,[1,*] and Berthold Göttgens[1,4,*]

[1]Wellcome-MRC Cambridge Stem Cell Institute, Department of Hematology, University of Cambridge, Cambridge, UK
[2]Hematological Cancer Genetics, Wellcome Trust Sanger Institute, Hinxton, UK
[3]York Biomedical Research Institute, Department of Biology, University of York, York, UK
[4]Lead contact
*Correspondence: nkw22@cam.ac.uk (N.K.W.), bg200@cam.ac.uk (B.G.)

## SUMMARY

Acute myeloid leukemia (AML) and myeloid neoplasms develop through acquisition of somatic mutations that confer mutation-specific fitness advantages to hematopoietic stem and progenitor cells. However, our understanding of mutational effects remains limited to the resolution attainable within immunophenotypically and clinically accessible bulk cell populations. To decipher heterogeneous cellular fitness to preleukemic mutational perturbations, we performed single-cell RNA sequencing of eight different mouse models with driver mutations of myeloid malignancies, generating 269,048 single-cell profiles. Our analysis infers mutation-driven perturbations in cell abundance, cellular lineage fate, cellular metabolism, and gene expression at the continuous resolution, pinpointing cell populations with transcriptional alterations associated with differentiation bias. We further develop an 11-gene scoring system (Stem11) on the basis of preleukemic transcriptional signatures that predicts AML patient outcomes. Our results demonstrate that a single-cell-resolution deep characterization of preleukemic biology has the potential to enhance our understanding of AML heterogeneity and inform more effective risk stratification strategies.

## INTRODUCTION

Acute myeloid leukemia (AML) is a heterogeneous hematological malignancy that arises from hematopoietic stem and progenitor cells (HSPCs). At least nine functional categories of genes are recurrently mutated and drive the heterogeneity in the process of leukemogenesis.[1] At the time of diagnosis, leukemic blasts are often present with multiple driver mutations with characteristic patterns of co-occurrence and mutual exclusivity.[1,2] These observations suggest that individual mutations possess unique and complementary functions that cooperate to promote leukemogenesis. Therefore, a deeper understanding of the distinct leukemogenic effects of individual mutations is a crucial next step in deciphering the intra- and inter-patient heterogeneity that results from different combinations of mutations.

Approximately 20% of AML cases are recognized as developing from antecedent myeloid neoplasms, such as myeloproliferative neoplasms and myelodysplastic syndrome,[3] following the acquisition of additional functional categories of mutations.[4,5] Recent studies on clonal hematopoiesis[6,7] have demonstrated that individual mutations confer varying degrees of fitness to HSPCs and modulate the risk for progression to leukemia. Thus, mutation-spe-

cific remodeling of hematopoietic clonal architecture plays a role from the earliest preleukemic stage through to full leukemic transformation. However, as early preleukemic changes are clinically silent, how single mutations differently perturb the entire hematopoietic system remains largely elusive.

Over the past half decade, single-cell technologies, including single-cell RNA sequencing (scRNA-seq), have reshaped our understanding of the hematopoietic hierarchy from a tree-like, stepwise model toward a differentiation landscape model.[8] More recent advances in computational and mathematical methods have further expanded the application of scRNA-seq beyond gene expression comparison, enabling the inference of more complex biological information, such as cellular fate probabilities[9] and metabolic activities.[10] As such, scRNA-seq provides a unique advantage in characterizing mutation-driven perturbations in various biological processes.

In this study, we perform scRNA-seq on bone marrow HSPCs and computationally infer the preleukemic changes in cell abundance, cellular lineage fate, metabolic activities, and gene expression across eight different mouse models harboring preleukemic mutations in *Jak2*, *Calr*, *Flt3*, *Npm1*, *Idh1*, *Dnmt3a*, *Ezh2*, and *Utx* (Figure 1A). Our preleukemic mouse cell atlas

**Cell Genomics**
**Resource**

**A** Data Integration

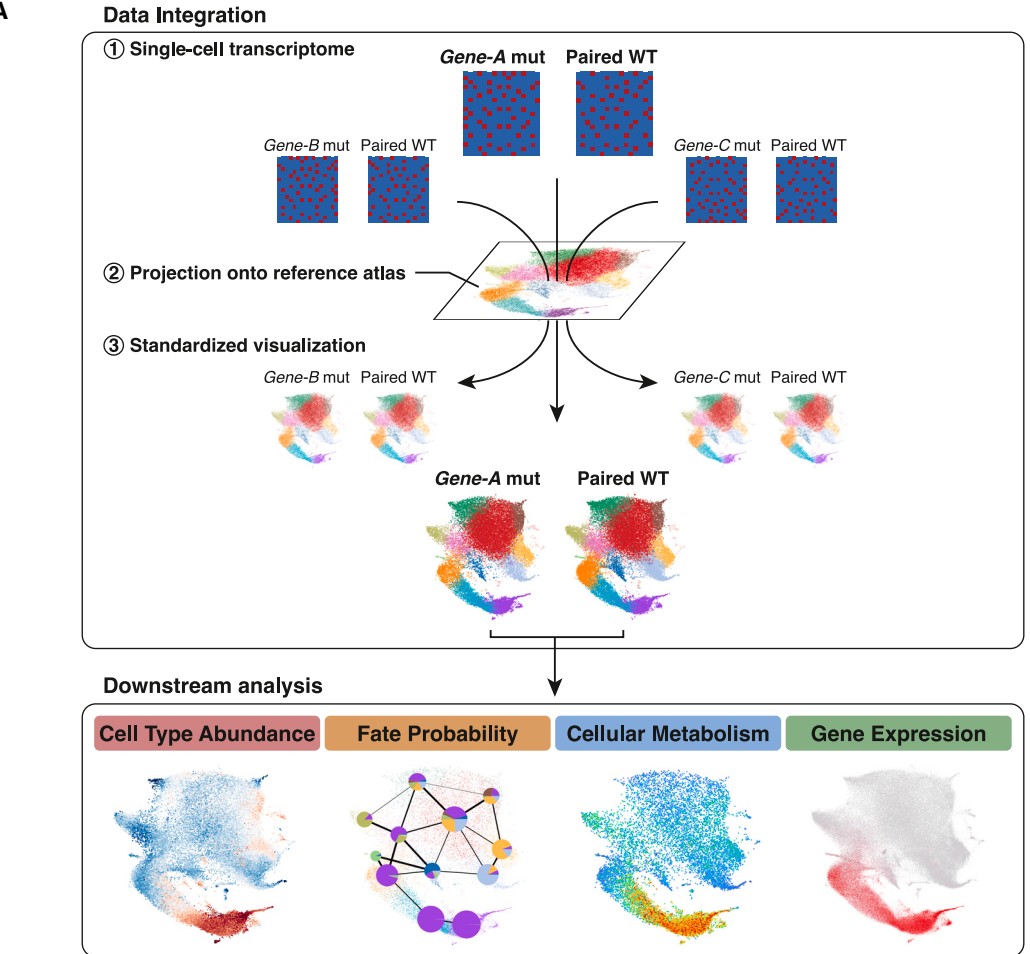

**B** Reference atlas
(44,802 cells)

**C** Preleukemic mouse cell atlas
(269,048 cells)

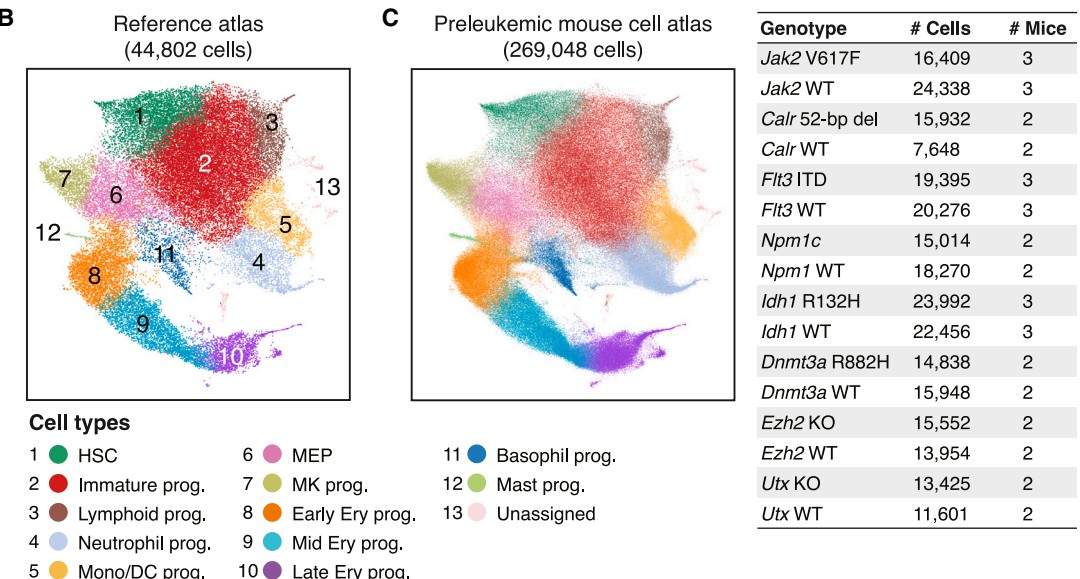

**Cell types**

| | | | | | | |
|---|---|---|---|---|---|---|
| 1 | HSC | 6 | MEP | 11 | Basophil prog. | |
| 2 | Immature prog. | 7 | MK prog. | 12 | Mast prog. | |
| 3 | Lymphoid prog. | 8 | Early Ery prog. | 13 | Unassigned | |
| 4 | Neutrophil prog. | 9 | Mid Ery prog. | | | |
| 5 | Mono/DC prog. | 10 | Late Ery prog. | | | |

| Genotype | # Cells | # Mice |
|---|---|---|
| *Jak2* V617F | 16,409 | 3 |
| *Jak2* WT | 24,338 | 3 |
| *Calr* 52-bp del | 15,932 | 2 |
| *Calr* WT | 7,648 | 2 |
| *Flt3* ITD | 19,395 | 3 |
| *Flt3* WT | 20,276 | 3 |
| *Npm1c* | 15,014 | 2 |
| *Npm1* WT | 18,270 | 2 |
| *Idh1* R132H | 23,992 | 3 |
| *Idh1* WT | 22,456 | 3 |
| *Dnmt3a* R882H | 14,838 | 2 |
| *Dnmt3a* WT | 15,948 | 2 |
| *Ezh2* KO | 15,552 | 2 |
| *Ezh2* WT | 13,954 | 2 |
| *Utx* KO | 13,425 | 2 |
| *Utx* WT | 11,601 | 2 |

*(legend on next page)*

(PMCA) can be explored using an interactive web portal at https://gottgens-lab.stemcells.cam.ac.uk/preleukemia_atlas/. This single-cell-resolution multi-scale analysis illustrates mutation-specific mechanisms of hematopoietic perturbation and identifies preleukemic genetic programs predictive of AML patient outcomes (Stem11 signature), thus establishing a novel framework for translating preleukemic biology into an improved treatment stratification strategy for AML patients.

## RESULTS

### Reference-based integration provides standardized visualization across multiple datasets

To unravel the mutation-specific preleukemic effects on hematopoietic differentiation, we performed scRNA-seq of lineage-negative/c-Kit-positive (LK) cells for 38 animals from eight different mutant mouse models and their wild-type littermates: homozygous *Jak2* V617F,[12] homozygous *Calr* 52 bp deletion,[13] heterozygous *Flt3*-ITD,[14] heterozygous *Npm1c*,[15] heterozygous *Idh1* R132H,[16] heterozygous *Dnmt3a* R882H,[17] homozygous *Ezh2* knockout (KO)[18] and homozygous *Utx* KO[19] (Table S1). A total of 269,048 single-cell transcriptomes passed stringent quality control measures (see STAR Methods), with a median of 2,819 genes detected per cell.

To minimize technical batch effects and enhance comparability between the mutant hematopoietic landscapes, we took advantage of our previously reported single-cell mouse hematopoietic atlas[11] (Figures 1B and S1A) and projected all mutant and wild-type samples onto this reference atlas with a reference-based integration and label transfer strategy using Seurat[20] (see STAR Methods). This enabled an automatic identification of the hematopoietic differentiation landscape within individual animals within a common uniform manifold approximation and projection (UMAP) space, while mitigating technical batch effects (Figure 1C). Our mutant animals did not develop overt leukemia at the time of analysis and consistently, the scRNA-seq landscapes revealed multi-lineage hematopoiesis in all mutant models (Figures S1B and S1C). Therefore, our data provide a unique view into the "preleukemic" window of various mutations, and our data integration strategy enables a standardized visualization across conditions to facilitate further quantitative comparisons of mutation-driven molecular, cellular, and tissue-scale alterations (Figure 1A).

### Preleukemic stem and progenitor cells show skewed abundance under steady state

To characterize the tissue-scale effects of individual mutations, we first performed differential cell abundance analysis. Using the Python package MELD,[21] we computed single-cell-resolution differential abundance by comparing the cell densities be-

tween paired mutant and wild-type samples (Figure S2A; see STAR Methods for details). When plotted in the common UMAP space, this differential abundance score (mutant relative likelihood) illustrated, at single-cell resolution, which transcriptionally defined cell types were increased or depleted in each mutant model (Figure 2A).

Reassuringly, this analysis was able to recapitulate the phenotype of *Jak2* V617F mice, in that in addition to a significant increase in later erythroid progenitors, a depletion of the most quiescent HSCs and megakaryocyte progenitors was detected (Figure S2B).[12] Of particular interest were the complex changes observed within the broadly defined hematopoietic stem cell (HSC) cluster of the *Flt3* ITD mutant animals (Figures 2B and 2C). To characterize these increased and depleted subpopulations of HSCs, a previously developed HSC score (molecular signature of long-term repopulating HSCs[22]) was calculated, which showed significantly higher scores in the depleted HSCs ($p = 1.3 \times 10^{-8}$; Figure 2D), indicating that the *Flt3* ITD mutation selectively depletes the most quiescent long-term HSCs. Importantly, cluster-wise comparison of population proportions failed to capture significant changes in the *Flt3* ITD HSCs (Figure 2E), as discrete population-level summarization loses information about intra-cluster heterogeneity. Moreover, although the bulk neutrophil progenitor population was significantly increased in the *Flt3* ITD mice ($p = 2.5 \times 10^{-3}$; Figure 2E), our single-cell-level evaluation revealed a significant depletion of later neutrophil progenitors (Figure 2F), suggesting a differentiation block at the late neutrophilic progenitor stage, which is consistent with the observation of terminal neutrophilic differentiation of *FLT3* ITD-positive AML blasts after treatment with *FLT3* inhibitors.[23,24]

Both young (12-week-old) and old (41-week-old) *Calr* mutant mice showed a consistent increase in HSCs and megakaryocyte progenitors, where the magnitude of changes was significantly greater in the old animal (Figures S2C and S2D). The *Utx* KO model showed a significant increase in neutrophil progenitors and megakaryocyte progenitors as well as a depletion of erythroid progenitors (Figure S2E), while the *Ezh2* KO model had only a few cells with significant changes (Figure S2E), presumably because *Ezh2* KO affects later B and T lymphopoiesis in adult hematopoiesis[25] and thus the impact on HSPC abundance is relatively small. The *Npm1c*, *Idh1* R132H, and *Dnmt3a* R882H mutations showed minimal changes in cellular abundance with the cluster median mutant relative likelihood falling within the range of 0.4–0.6 for all cell types (Figure 2A) and with no statistically significant differential abundance, indicating that these mutations contribute to leukemogenesis in a way that does not involve dramatic changes in tissue-wide population abundance at the HSPC level. Altogether, our differential abundance analysis has provided a mutation-specific tissue-wide picture of skewed hematopoiesis at single-cell resolution.

---

**Figure 1. Standardized visualization across multiple mutant landscapes using reference-based data integration**

(A) Schematic overview of the analysis workflow in this study. Data integration: scRNA-seq data of individual animals are first projected onto the reference mouse hematopoietic atlas and all visual representations of the results are shown in the common UMAP space. Downstream analysis: the downstream analysis modules then infer differential abundance, cellular fate probability, cellular metabolic activity and gene expression changes. All results are shown in the common UMAP space to permit comparison among genotypes.

(B) UMAP plot of the reference mouse hematopoietic atlas[11] (44,802 cells). HSC, hematopoietic stem cell; prog, progenitors; Mono, monocyte; DC, dendritic cell; MEP, megakaryocyte-erythroid progenitors; MK, megakaryocyte; Ery, erythroid.

(C) Integrated preleukemic mouse hematopoietic atlas (38 animals, 269,048 cells). Cells are color coded according to cell types as in (B). WT, wild type; KO, knockout.

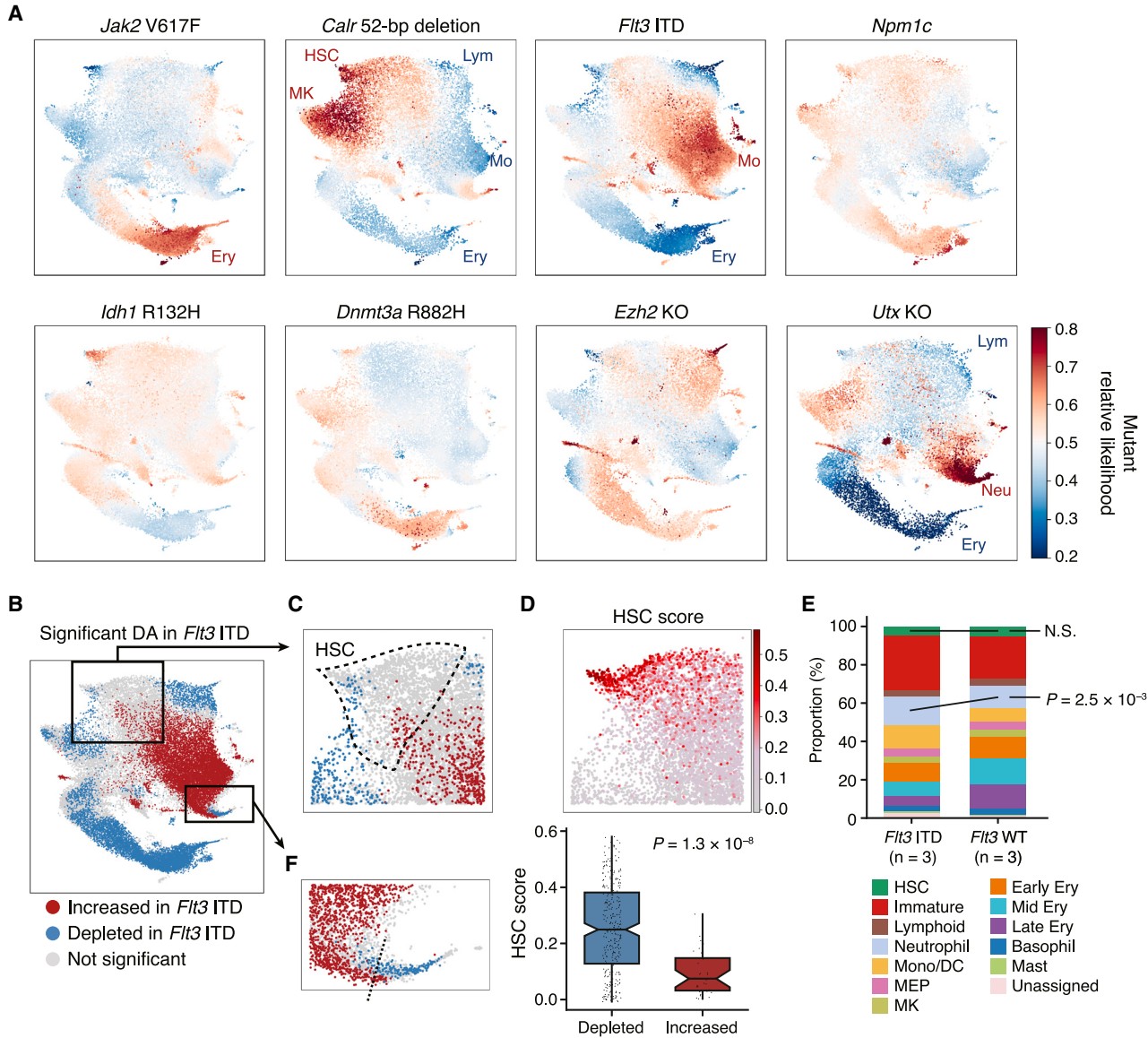

**Figure 2. Depicting tissue-scale subpopulation abundance at single-cell resolution**

(A) Differential abundance landscapes of eight preleukemic mutant models. Higher likelihood (red) corresponds to higher abundance and lower likelihood (blue) to lower abundance in each mutant model compared with paired wild-type samples. All results are presented in the same color scale ranging from 0.2 to 0.8 for comparability. Cell types with a median mutant relative likelihood of >0.6 or <0.4 are indicated.

(B) Statistically significant differential abundance (DA) in the *Flt3* ITD model. p values were derived using a two-sided t test. Cells with raw p values < 0.05 and BH-adjusted p values < 0.25 are colored red (abundant) or blue (depleted in the *Flt3* ITD model).

(C) A magnified section of the HSC cluster region. The boundary of the HSC cluster is outlined with dashed lines.

(D) Top: the HSC score in the *Flt3* ITD and wild-type HSCs. Bottom: significant difference in the HSC score between *Flt3* ITD-depleted (mutant relative likelihood < 0.4) and increased (mutant relative likelihood > 0.6) HSCs. Boxplots show median and first/third quartiles. The whisker extends from the smallest to the largest values within 1.5 × IQR from the box hinges. The p value is from a two-sided Wilcoxon rank-sum test.

(E) Stacked bar plots showing the cell type proportions in the *Flt3* ITD and wild-type animals. Cell types are color coded as in Figure 1B. Statistical significance was determined with two-sided t test.

(F) A magnified section of the neutrophil progenitor cluster region. The boundary between significantly expanded neutrophil progenitors (red) and depleted later neutrophil progenitors (blue) is indicated by a dashed line.

## Single-cell transcriptome deciphers mutation-driven fate bias in early HSPCs

Next, to explore whether and how each mutation alters the cell-intrinsic potential of differentiation, we inferred cellular fate prob-abilities using CellRank.[9] On the basis of a cell-to-cell neighbor-hood graph and diffusion pseudotime,[26] cell-to-cell transition probabilities were first computed, and then the fate probabilities toward seven major hematopoietic lineages (megakaryocyte,

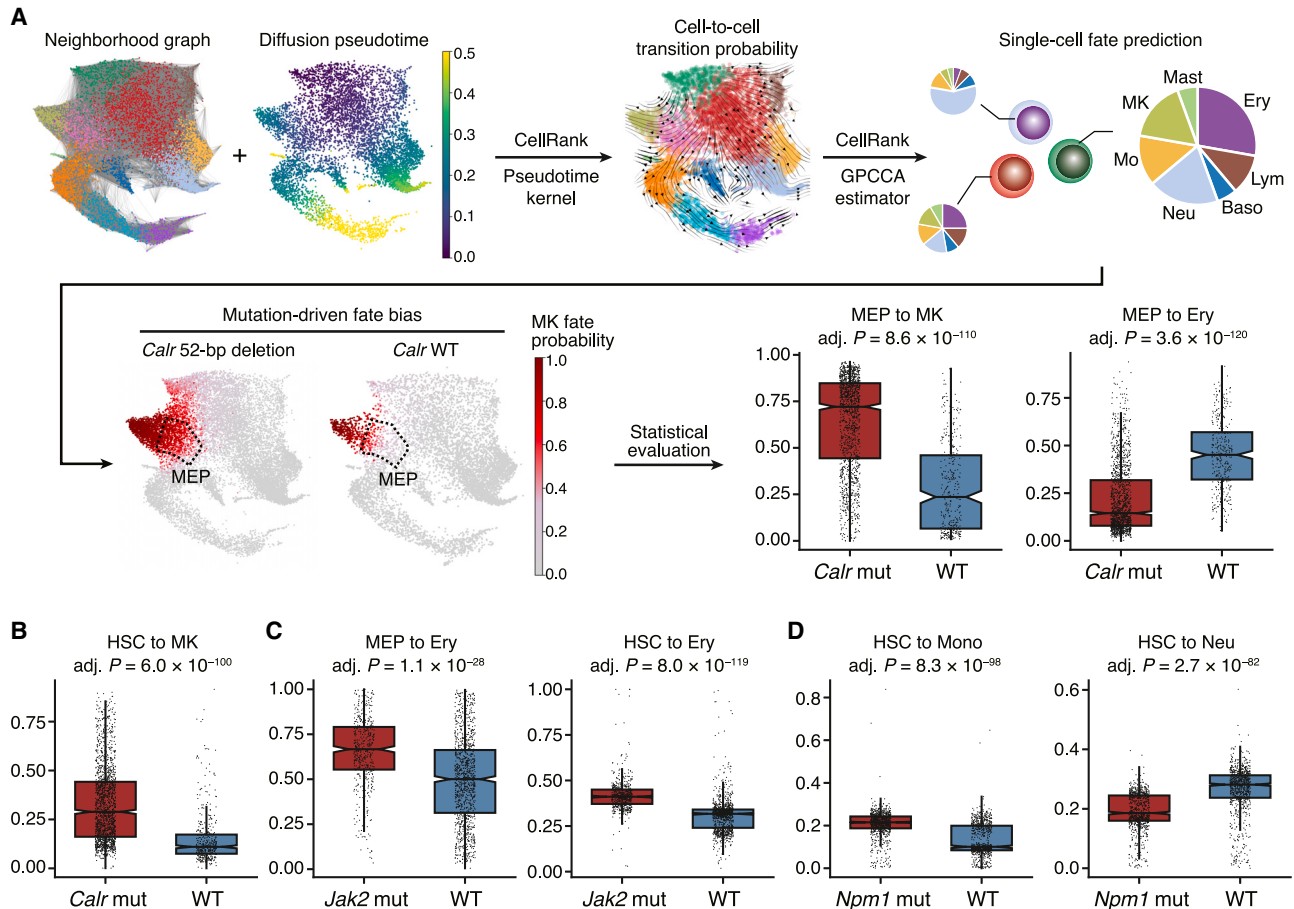

**Figure 3. Mutation-driven fate bias in early HSPCs**

(A) Workflow of CellRank-based differential fate probability analysis. A neighborhood graph and diffusion pseudotime were computed (top left) and used to infer cell-to-cell transition probabilities (top middle). On the basis of the transition matrix, the fate probabilities toward seven hematopoietic lineages (megakaryocyte, erythroid, lymphoid, neutrophil, monocyte, basophil, and mast cell) were estimated for the individual cells (top right). The single-cell estimates of fate probabilities were then compared between the paired mutant and wild-type samples (bottom). Ery, erythroid; Lym, lymphoid; Baso, basophil; Neu, neutrophil; Mo, monocyte; MK, megakaryocyte.

(B) Significant difference in the megakaryocyte probability between *Calr* mutant and wild-type HSCs.

(C) Significant differences in the erythroid probability between *Jak2* mutant and wild-type MEPs (left) and HSCs (right).

(D) Significant differences in the monocyte (left) and neutrophil (right) probability between *Npm1* mutant and wild-type HSCs. Boxplots show median and first/third quartiles. The whisker extends from the smallest to the largest values within 1.5 × IQR from the box hinges. p values are from logistic regression and likelihood ratio test and are BH adjusted.

erythroid, lymphoid, neutrophil, monocyte, basophil, and mast cell) were computed for each cell, allowing visual and statistical evaluation of the fate bias in each mutant model (Figures 3A and S3).

Of note, cell fate probability analysis identified a significant increase in megakaryocytic and a decrease in erythroid fate probability in the *Calr* mutant megakaryocyte-erythroid progenitors (MEPs; Figure 3A), consistent with the higher and lower production of megakaryocyte and erythroid progenitors, respectively (Figure 2A). Intriguingly, megakaryocytic fate bias was detectable even at the level of HSCs (Figure 3B), although the effect was smaller than in MEPs (median fate difference of 17.9% in HSCs (28.9% in *Calr* mutant vs. 11.0% in wild type) compared with 48.5% in MEPs (72.1% in *Calr* mutant vs. 23.6% in wild type). Similarly, erythroid fate bias was detected in the *Jak2*

V617F HSCs and MEPs with a greater difference in MEPs (median fate difference of 9.4% in HSCs (41.1% in *Jak2* mutant vs. 31.7% in wild type) compared with 16.6% in MEPs (66.7% in *Jak2* mutant vs. 50.0% in wild type); Figure 3C). As the megakaryocytic and erythroid bias in the *Calr* and *Jak2* mutated HSCs has been previously demonstrated experimentally,[27,28] our results not only computationally recapitulated these phenotypes but also provided a quantitative view of how hematopoietic lineage fates are progressively biased from HSCs to more differentiated oligo-/bi-potent progenitors. Other lineage bias detected in mutant HSCs included monocytic bias by the *Flt3* ITD and *Npm1c* mutations, lymphoid bias by the *Ezh2* KO and neutrophilic and megakaryocytic bias by the *Utx* KO (Table S2). Of note, *Npm1c* mutant HSCs showed monocytic bias at the expense of neutrophilic fate (Figure 3D), consistent with the

highest incidence of monocytic AML (FAB M4/M5) in the *NPM1*-mutated patients.[29] No positive lineage bias was detected in the *Dnmt3a* and *Idh1* mutant models (Table S2). Overall, our fate probability analysis provided insights into how different mutations may alter HSC fates and have the potential to tilt the balance of hematopoietic lineages.

### Neural network modeling reveals distinct metabolic alterations associated with different mutations

As the differential abundance and fate probability analyses have revealed tissue and cellular scale perturbations, we next sought to characterize the molecular level alterations in the preleukemic conditions. For this purpose, we computationally inferred the cellular metabolic activities from scRNA-seq data using a deep neural network model of the Python package scFEA[10] and investigated mutation-driven metabolic alterations (Figure 4A).

By comparing the paired mutant and wild-type samples, we first asked whether our results recapitulate known metabolic changes in mutant mouse hematopoiesis. MEPs from *Jak2* V617F mutant mice are known to upregulate glycolysis, the tricarboxylic acid (TCA) cycle and nucleotide synthesis pathways to fuel their increased energy requirements for overproduction of erythrocytes.[30] Consistently, neural-network-based metabolic profiling successfully identified a significant upregulation of these pathways in MEPs within our *Jak2* V617F model (Figures S4A and S4B). Moreover, a recent report has shown by isotope tracing that the co-occurrence of *Ezh2* KO and *Nras* G12D mutation enhances the branched-chain amino acid (BCAA) metabolism while *Ezh2* KO or *Nras* G12D mutation alone does not alter the activity of this metabolic pathway.[31] Again, our metabolic profiling using the scRNA-seq data from the same mouse models[32] successfully identified the enhanced BCAA metabolism only in the double-mutant model (Figures S4C and S4D). These results thus verified the applicability of this method for the hematopoietic system.

Encouraged by the validation results, we further explored the metabolic alterations in our mutant models. As glycolysis and the TCA cycle are the major source of cellular energy, we asked whether the increased cell populations in each mutant model (Figure 2A) are accompanied by increased energy generation. Indeed, the *Jak2* and *Calr* mutations (group 1 mutations) showed global upregulation of these metabolic pathways in the expanded erythroid and megakaryocyte progenitors, respectively (Figure 4B). This is consistent with the constitutive activation of the JAK-STAT signaling pathway and cellular proliferation by these mutations,[33,34] which augments cellular energy requirements. In contrast, the mutations in the epigenetic regulators *Idh1*, *Ezh2*, and *Utx*, as well as the *Npm1* mutation (group 2 mutations), showed significant downregulation of these energy-generating pathways in their respective targets of cell type accumulation (Figures 4C and S4E), suggesting that accumulation of cells was the consequence of a differentiation block rather than a proliferative push.

The *Dnmt3a* and *Flt3* ITD mutations did not fit into either group; the *Dnmt3a* mutation did not show significant metabolic changes in these energy-producing pathways (Figure 4D). Interestingly, the *Flt3* ITD mutation induced downregulation of the TCA cycle while maintaining or upregulating the glycolysis reactions (Fig-ure 4E). This dissociation between glycolysis and the TCA cycle is consistent with the Warburg effect, a cancerous metabolic alteration in favor of glucose metabolism into lactate rather than harnessing the TCA cycle.[35] Altogether, our single-cell metabolic analysis has successfully recapitulated known mutation-driven metabolic alterations in the hematopoietic system and further proposed a classification of leukemogenic mutations on the basis of the potential metabolic modes of action.

### Leukemogenic mutations induce cell-type-specific and dynamic changes in gene expression

To further characterize the preleukemic perturbations, we next sought to identify transcriptional signatures associated with the lineage bias in each mutant model. To this end, we performed cell-type-wise differential expression analysis of the paired mutant and wild-type samples. This revealed highly variable numbers of genes differentially expressed in different cell types and in different mutant models (Figure 5A). Reassuringly, the changes in gene expression seen in the mouse models recapitulated the gene expression signatures seen in patients with the respective mutation[1,36–39] (Figure S5A). Assuming the number of differentially expressed genes as a proxy for the perturbing effect size, the *Calr* deletion mutation, *Ezh2* KO and *Utx* KO showed the largest perturbing effects with >500 genes differentially expressed in one or more cell types. In contrast, the *Idh1* R132H mutation induced <10 differentially expressed genes in any cell types in our model. This is consistent with a previous expression microarray analysis of an *Idh1* R132H model,[40] which identified no significantly differentially expressed genes with false discovery rates < 0.05, suggesting non-transcriptomic mechanisms of the *Idh1* R132H mutation in leukemogenesis.

Of note, in the *Jak2* V617F model, early erythroid progenitors showed the greatest transcriptomic change (Figure 5A, top left panel) with a significant activation of cell cycle regulators (Figure 5B), while population-level abundance was increased only in later erythroid progenitors (Figure 2A). Likewise, MEPs in the *Calr* mutant model and immature progenitors in the *Flt3* ITD model showed a greater number of differentially expressed genes than megakaryocyte progenitors and neutrophil/monocyte progenitors, respectively (Figure 5A), suggesting that transcriptomic alterations precede the expansion of later progenitor stages. Furthermore, consistent with the increase in erythroid fate probability in the *Jak2* V617F MEPs (Figure 3C), the regulatory genes of megakaryocyte differentiation were significantly downregulated in the *Jak2* V617F MEPs (Figure 5C). Collectively, these results indicate that mutations induce differentiation stage-specific transcriptomic programs that eventually lead to a tissue-scale hematopoietic skew.

To better capture stage-specific dynamic expression changes, we next modeled gene expression as a function of pseudotime and compared the pseudotemporal expression patterns using a generalized additive model.[41] By defining lineage trajectories on the basis of lineage fate probability (see STAR Methods), we first compared the pseudotemporal gene expression patterns between the *Jak2* V617F and wild-type erythroid trajectory (Figure S6A) and identified 80 genes with significantly altered expression patterns (Table S3). Among them, a megakaryocytic differentiation marker *Pf4* showed lower activation in the pseudotime

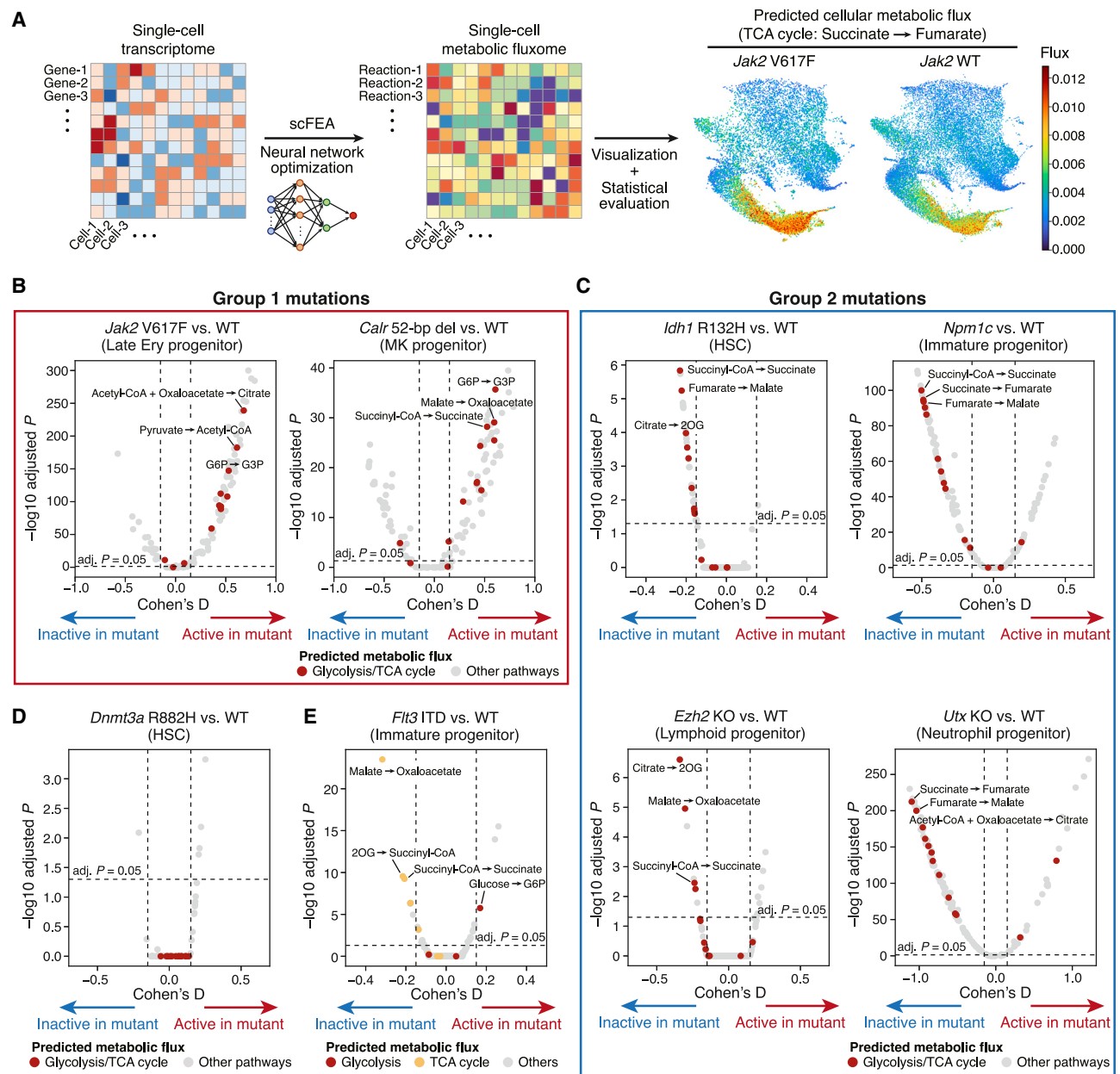

**Figure 4. Transcriptome-based metabolic profiling reveals distinct metabolic consequences of different mutations**

(A) Workflow of neural-network-based metabolic profiling. Using the expression levels of enzyme genes as input, a deep neural network model was optimized to estimate the activities of 168 metabolic reactions in the individual cells. The cellular metabolic estimates were then used for statistical comparisons.

(B–E) Volcano plots comparing glycolysis and TCA cycle activities in the mouse models of group 1 mutations (*Jak2* and *Calr*) (B), group 2 mutations (*Idh1*, *Npm1*, *Ezh2*, and *Utx*) (C), *Dnmt3a* mutation (D), and *Flt3* ITD (E). The x and y axes represent Cohen's D standardized difference of means and $-\log_{10}$ adjusted p values, respectively. Each dot represents each metabolic reaction module and is colored according to the functional pathways. The horizontal dotted line indicates the adjusted p value of 0.05; the vertical dotted lines indicate the Cohen's D values of −0.15 and 0.15.

window corresponding to the MEP stage (Figure 5D), while globin genes (*Hba-a1*, *Hba-a2*, *Hbb-bs*, and *Hbb-bt*) showed an earlier onset of expression from the mid-erythroid progenitor stage (Figure 5E; Table S3). *Tfrc* (encoding the transferrin receptor or CD71) as well as cell cycle regulators showed constant upregulation throughout the erythroid differentiation (Figure 5E). These results indicate that not a single factor but a combination of (1) lower activation of megakaryocytic differentiation regulators, (2) early activation of erythroid differentiation genes and (3) constant upregulation of cell cycle regulators is the transcriptomic underpinning for the erythroid bias observed within the *Jak2* V617F-homozygous HSPCs.[12] Early activation of megakaryocytic and myelomonocytic differentiation markers were also observed in the *Calr* mutant megakaryocytic trajectory and the *Flt3*-mutant

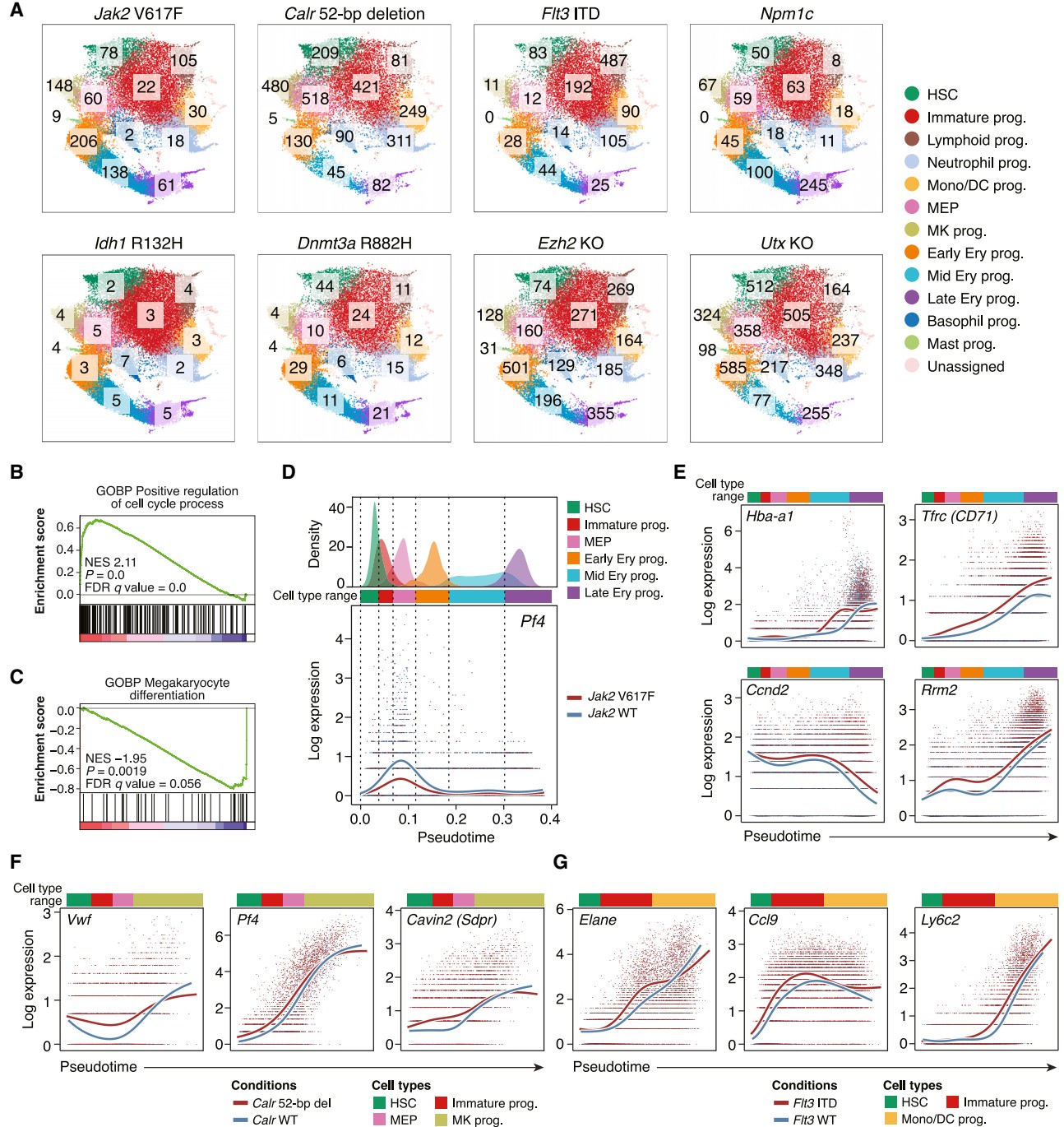

**Figure 5. Perturbed regulation of gene expression in preleukemic mutant models**

(A) The number of genes differentially expressed in each cell type from each mutant model compared with the wild-type counterpart. The numbers are shown over the corresponding cell types in the individual UMAP plots. HSC, hematopoietic stem cell; prog, progenitors; Mono, monocyte; DC, dendritic cell; MEP, megakaryocyte-erythroid progenitors; MK, megakaryocyte; Ery, erythroid.

(B) Significant upregulation of cell cycle regulators in *Jak2* mutant early erythroid progenitors. Gene Ontology (GO) terms for biological processes (BPs) were evaluated. NES, normalized enrichment score; FDR, false discovery rate.

(C) Significant downregulation of genes regulating megakaryocytic differentiation in *Jak2* mutant MEPs.

(D) Differential expression dynamics of *Pf4* between the erythroid trajectory of the *Jak2* mutant and the paired wild-type samples. The upper panel shows the pseudotime distribution of each cell type, defining the pseudotime ranges of dominant cell types. The lower panel shows the pseudotemporal expression patterns

*(legend continued on next page)*

myelomonocytic trajectory, respectively (Figures 5F, 5G, S6B, and S6C; Table S3). Moreover, the unfolded protein response genes, reported to be upregulated in *CALR*-mutated myeloproliferative neoplasms,[37] were significantly upregulated in our *Calr* mutant model with pseudotemporal expression patterns similar to those in patients with *CALR*-mutated essential thrombocythemia[37] (Figures S5B–S5F). Overall, our differential expression analysis has enabled quantification of the perturbing effects of different mutations and revealed the putative transcriptomic basis of lineage bias.

### PMCA pipeline streamlines single-cell analysis of preleukemic mouse models

To test the external applicability of our analysis pipeline, we next analyzed a previously published scRNA-seq dataset of a *Tet2* KO mouse model.[42] The hematopoietic differentiation landscape was automatically identified (Figure S7A), and differential abundance and fate probability analyses illustrated expansion of HSCs and myeloid-biased hematopoiesis within the *Tet2* KO model (Figures S7B and S7C), consistent with previous observations.[43–45] Furthermore, single-cell metabolic analysis demonstrated reduced activity of glycolysis and the TCA cycle (Figure S7D), classifying the *Tet2* mutation as a group 2 mutation. Differential expression analysis identified the largest transcriptomic changes in the immature to myeloid progenitors (Figure S7E), and myeloid differentiation/maturation markers showed significantly altered expression patterns along the neutrophilic differentiation trajectory (Figures S7F and S7G), in concordance with impaired late neutrophilic maturation by the *Tet2* mutation.[46] As loss-of-function mutations in the *TET2* gene are among the most common drivers of human preleukemia,[47–49] this analysis not only provides complementary results to the original publication but also serves to demonstrate the broad utility of our PMCA pipeline for streamlined characterization of mutational effects.

### Preleukemic lineage perturbation signature defines patients with the most immature and refractory AML

Finally, in order to evaluate the translational applicability of our analyses, we asked whether the molecular signatures derived from our pseudotemporal differential expression analysis can identify clinically relevant patient characteristics. As our analysis and previous reports[12,13,50] indicate that the *Jak2*, *Calr*, and *Flt3* mutations actively drive proliferation and biased differentiation, we combined the erythroid, megakaryocytic, and myelomonocytic bias signatures dysregulated by these mutations (Table S3). This allowed us to develop a comprehensive gene signature representing the biased differentiation toward the major myeloid lineages (hereafter, preleukemic lineage perturbation signature [PLPS]; 216 genes). We then applied this signature to the clustering of gene expression data from AML patients (Figure 6A; Table S4). The PLPS genes effectively separated the TCGA (The Cancer Genome Atlas) AML patient samples[1] according to

the morphological FAB classification in the principal component space (Figure 6B). Consistent with this, hierarchical clustering on the basis of the PLPS genes identified four clusters (Stem, Intermediate, Monocytic, and Granulocytic) with distinct expression patterns of differentiation marker genes (Figures 6C, S8A, and S8B). The stem-cell-like immature transcriptomic status of the Stem cluster was further confirmed by significant enrichment of the previously developed LSC17 score[51] (Figures S8C and S8D). Notably, these four PLPS-based clusters showed significant differences in overall survival, with the Stem cluster having the poorest survival rate (Figure 6D).

To further refine the PLPS genes and pinpoint genes associated with poor prognosis, we compared the expression of PLPS genes between the Stem cluster and the other three clusters. Of the 216 PLPS genes, twelve were significantly upregulated in the Stem cluster (Benjamini-Hochberg [BH]-adjusted p < 0.05 and log$_2$ fold change > 2; Figures 6E and S9). To assess the predictive capability of these 12 genes for patient prognosis, we next analyzed the Beat AML cohort,[2] in which the PLPS genes also partitioned the patient samples according to their differentiation status (Figures S10A and S10B). Of the 12 genes upregulated in the Stem cluster of the TCGA cohort, eleven were also expressed in the Beat AML cohort (hereafter, Stem11; Figure 6E). The sum of the expression *Z* scores of the Stem11 genes (Stem11 score) showed a bimodal distribution in which the top 15th percentile formed a peak with higher Stem11 scores (Stem11-high patients; Figure S10C). Importantly, the Stem11-high patients exhibited significantly poorer overall survival than the Stem11-low patients (p = 1.2 × 10$^{-3}$; Figure 6F), indicating the predictive power of the Stem11 system. Despite the presence of physiological differentiation markers for megakaryocytic (*CAVIN2*, *VWF*, and *PF4*), erythroid (*CA1*) and lymphoid (*DNTT*) lineages within the Stem11 genes, the blast content of the sequenced samples did not correlate with the Stem11 score (Figure S10D), suggesting that the higher Stem11 gene expression is due to a specific expression program preferentially seen in patients with poor prognosis.

To further ascertain how the Stem11 score stratifies patient prognosis, we next evaluated the association between the Stem11 score and clinical risk factors. Although the Stem11 score did not correlate with patient age (Figure S10E), Stem11-high patients were significantly more likely to belong to the ELN2017 adverse risk group[52] (Figure 6G). Specifically, 27 of 30 (90%) Stem11-high patients belonged to the ELN2017 adverse risk group, almost halving the group (Figure 6G). Additionally, the Stem11 score was significantly associated with overall survival even in the ELN2017 adverse risk group (Figure 6H). This is in contrast to the LSC17 system,[51] which broadly divided the entire cohort into two prognostic groups but did not segregate the ELN2017 adverse risk group (Figures S10F and S10G). As previously reported in another independent cohort,[53] the ELN2017 intermediate and adverse risk groups did not differ in patient outcomes in the Beat AML cohort (Figure S10H);

---

of *Pf4*. The red (*Jak2* V617F) and blue (wild-type) lines show the expression smoothers estimated by a negative binomial generalized additive model.[41] Each dot shows the log-normalized expression and the pseudotime of each cell.

(E–G) Significantly altered gene expression patterns in the *Jak2* mutant and wild-type erythroid trajectory (E), the *Calr* mutant and wild-type megakaryocyte trajectory (F), and the *Flt3* mutant and wild-type myelomonocytic trajectory (G). The pseudotime ranges of dominant cell types are indicated with colored bars.

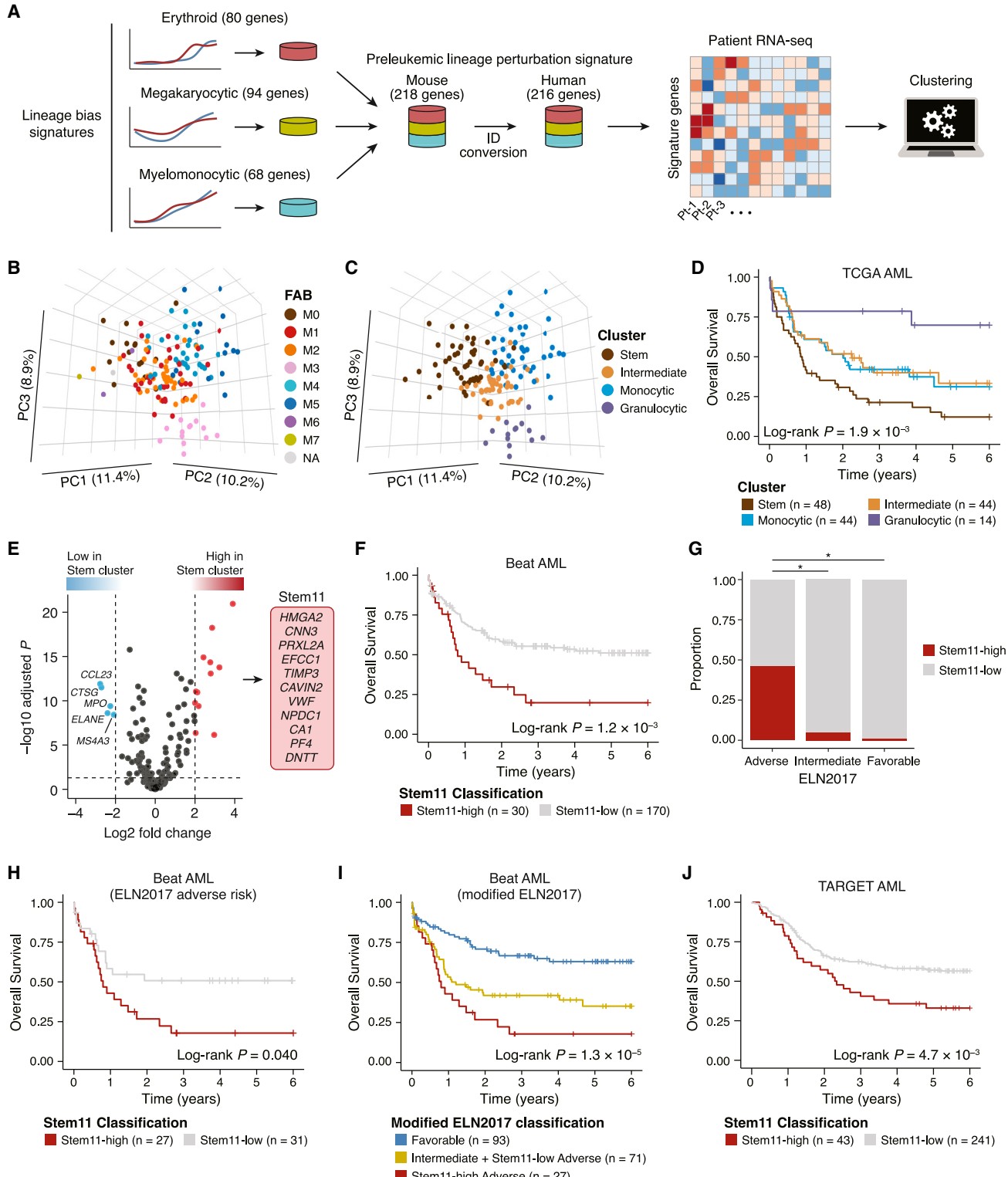

**Figure 6. Prognostic relevance of preleukemic lineage perturbation signature genes**

(A) Overview of the analysis of patient RNA-seq data. The erythroid, megakaryocytic, and myelomonocytic bias signatures derived from the pseudotemporal differential expression analysis were combined to develop the PLPS (preleukemic lineage perturbation signature) genes, which were then used to cluster the patient data.

*(legend continued on next page)*

however, re-stratification of the Stem11-low, ELN2017 adverse-risk patients to the intermediate risk group enabled better stratification of patient prognosis (Figure 6I). Importantly, all cases with *KMT2A* rearrangements had low Stem11 scores (Figure S11A), suggesting that re-stratification of this group to the intermediate risk group may be beneficial. Moreover, among the adverse risk mutations in the recently updated ELN2022 classification,[54] *TP53*, *RUNX1*, *BCOR*, *SF3B1*, and *ZRSR2* mutations were significantly more prevalent among the most refractory Stem11-high patients (Figure S11A), indicating that these mutations maintain characteristics inherited from preleukemic HSPCs. When applied to a cohort of AML patients uniformly treated with intensive chemotherapy,[55] the Stem11 classification showed a significant association with overall survival in both univariate ($p = 7.2 \times 10^{-3}$) and multivariate analysis ($p = 7.5 \times 10^{-3}$), thus demonstrating prognostic value independent from both patient age and ELN2017 cytogenetic/molecular risk classification (Figures S10I and S10J). The Stem11 score was also significantly associated with patient outcomes in the TARGET pediatric AML cohort[56] ($p = 4.7 \times 10^{-3}$; Figure 6J). As pediatric and adult AML have distinct mutational profiles, the ELN2022 adverse risk mutations developed for adult patients were rare and not significantly enriched among the Stem11-high pediatric patients, while the Stem11-high pediatric patients had significantly higher frequencies of *PTPN11* and *WT1* mutations (Figure S11B). These results indicate that the PLPS and Stem11 signatures identified through our integrated single-cell analysis of preleukemic hematopoiesis can define patients with the most immature and refractory AML for both adult and pediatric cohorts, regardless of their distinct genetic basis, thus providing improved molecular risk stratification.

## DISCUSSION

Premalignant biology is an emerging focus of global research, with the potential to advance our understanding of cancer development and to guide effective detection and intervention strategies for cancers.[57] In this study, we present a scRNA-seq-based multi-scale analysis framework for characterizing mutation-driven preleukemic perturbations and demonstrate that the molecular signatures of preleukemic lineage perturbations decipher AML heterogeneity and suggest improved risk stratification strategies.

The initial step of reference-based single-cell data integration systematically anchors new datasets within a hematopoietic differentiation landscape, providing a standardized visualization that facilitates comparison across different perturbation models.

Importantly, our characterization of mutation-specific perturbations is conducted by comparing paired mutant and wild-type samples with matched age, sex, genetic background, and pIpC regimens (for the *Npm1*, *Idh1*, *Dnmt3a*, *Ezh2*, *Utx*, and *Calr* [41-week-old] models). In addition, different experimental batches (i.e., different sample collections and library preparations) are included as covariates in the statistical models and adjusted to compute statistical significance. In line with the growing recognition of the importance of accounting for biological and technical variations in single-cell comparative analysis,[58,59] our experimental and statistical design enables the robust identification of true mutation-driven perturbations while mitigating non-pathological false positives.

Biologically, the four downstream analysis modules collectively illustrate how individual mutations drive their target molecular programs (i.e., metabolic and expression), bias cellular lineage fates and ultimately skew the tissue-wide landscapes of hematopoiesis. Here, the fate probability analysis and metabolic flux analysis exemplify the transformative capabilities of scRNA-seq beyond gene expression comparisons. On the basis of the recently developed machine learning models,[9,10] these analyses successfully recapitulate previously demonstrated experimental observations[27,28,30,31] (Figures 3B, 3C, and S4), thereby validating their applicability in the study of hematopoiesis. Notably, our differential metabolic flux analysis identified two groups of mutations with opposing impacts on glycolysis and TCA cycle reactions. As these energy-generating pathways are crucial for cell proliferation and differentiation,[60] our results suggest that the group 1 mutations (*Jak2* and *Calr*) skew hematopoiesis through active proliferation and biased differentiation, while the group 2 mutations (*Idh1*, *Npm1*, *Ezh2*, and *Utx*) lead to reduced energy demands and passive accumulation of specific cell lineages. This is specifically highlighted by the contrasting effects on cell cycle status seen in the *Calr* mutation and *Utx* KO models, whereby both have increased abundance of megakaryocyte progenitors (Figure 2A), yet the *Calr* mutation led to an increased proportion of cycling $G_2$/M-phase cells, while the *Utx* KO resulted in an increase in $G_1$-phase cells (Figure S4F). Interestingly, all the group 2 mutations are known to act through epigenetic dysregulation.[19,40,61,62] As epigenetic processes play a crucial role in normal and malignant stem cell differentiation,[63] our results suggest that downregulation of energy-generating metabolic pathways may represent at least part of the metabolic underpinning of the differentiation block caused by the group 2 mutations. Metabolic modulation is a promising therapeutic strategy for targeting cancer cell-specific dependencies by limiting cell growth or inducing differentiation.[64,65] Thus, our

(B and C) Principal-component analysis of the TCGA cohort. Samples are colored according to the FAB classification (B) and the patient clusters (C). PC1, PC2, and PC3 represent the first three principal components, with the percentage of variance explained indicated on the axes.

(D) Survival analysis comparing the overall survival of the different clusters of TCGA AML cohort.

(E) Volcano plot showing the differential expression of PLPS genes. Genes with significant upregulation (BH-adjusted $p < 0.05$ and $\log_2$ fold change > 2; red) and downregulation (BH-adjusted $p < 0.05$ and $\log_2$ fold change < −2; blue) are color coded.

(F) Survival analysis comparing the Stem11-high and Stem11-low groups in the Beat AML cohort.

(G) Proportions of Stem11-high patients in each of the ELN2017 risk groups. p values are from Fisher's exact test. *$p < 1.0 \times 10^{-5}$.

(H) Survival analysis comparing the Stem11-high and Stem11-low patients among the ELN2017 adverse risk group in the Beat AML cohort.

(I) Survival analysis comparing our modified ELN2017 risk groups, where the Stem11-low, ELN2017 adverse-risk patients were re-stratified to the intermediate risk group.

(J) Survival analysis comparing the Stem11-high and Stem11-low groups in the TARGET AML cohort.

single-cell metabolic profiling provides novel insights into possible early metabolic intervention by uncovering shared and unique metabolic alterations in preleukemic conditions.

The differential abundance and differential expression analysis modules further highlight the unique advantages of using scRNA-seq in characterizing perturbations. By comparing cell densities in the reference-based common latent space, our differential abundance analysis enables statistical evaluation of mutant cell expansion and depletion at the single-cell resolution. As this approach is not constrained by predefined cell type boundaries, it can pinpoint perturbed cell populations with a level of resolution that is unattainable through discrete cluster-wise comparison or immunophenotype-based quantification, as exemplified by our identification of a late neutrophilic differentiation block by the *Flt3*-ITD mutation. Likewise, the continuous resolution of gene expression analysis allows for the identification of genes with perturbed dynamic expression patterns, as opposed to discrete comparison of averaged expression in transcriptionally or immunophenotypically defined cell types. Consequently, we show that individual mutations orchestrate diverse transcriptional programs over the course of differentiation, including (1) early activation of specific lineage markers, (2) lowering expression of regulators of alternative lineages, and (3) modulation of cell cycle regulators, collectively perturbing hematopoietic differentiation.

Derived from the preleukemic molecular signatures, our PLPS and Stem11 gene sets are likely to at least in part, signify aberrant lineage priming in mutant HSPCs. Intriguingly, the PLPS genes effectively distinguish the heterogeneous differentiation status of clinical AML samples, suggesting that perturbed lineage priming during the preleukemic phase determines the eventual stages of arrested differentiation in AML blasts. Furthermore, the Stem11 score, derived from a subset of PLPS genes overexpressed in the most immature AML cases, identifies both adult and pediatric AML patients with the poorest survival outcomes. Of the 11 genes, *HMGA2* has been implicated in poorer prognosis in various human cancers including AML.[66] Otherwise, the Stem11 system has no gene overlap with previously published prognostic systems for AML,[51,67,68] thus representing a unique feature of refractory AML reflecting preleukemic perturbations. Considering the presence of multiple lineage markers (e.g., *DNTT*, *PF4*, and *CA1*) in the Stem11 genes, Stem11-high AML may capture features reminiscent of acute leukemia of ambiguous lineage, which is associated with immature cells of origin and inferior prognosis.[69] For adult patients, the Stem11 system could have clinical impact by recommending less toxic treatment options without hematopoietic cell transplantation (HCT) for more patients by re-stratifying the Stem11-low, ELN2017 adverse-risk patients to the intermediate risk group. In pediatric cases, HCT is usually reserved for the highest risk children and relapsed cases to balance long-term toxicity against survival risk; nevertheless, molecular genetic risk factors to stratify treatment are not well established and are thus urgently needed.[70] Therefore, in the pediatric context, the Stem11 system helps identify patients for whom toxic but curative HCT would be indicated.

Finally, in the present study, we have characterized mutations relevant to varying degrees of preleukemic perturbation and

different clinical contexts; *DNMT3A* and *TET2* mutations are the most common drivers of clonal hematopoiesis,[47–49] while *JAK2* and *CALR* mutations are rather specific to myeloproliferative neoplasms and secondary AML.[2,71] *IDH1*, *EZH2*, and *UTX* mutations are known to be rare drivers of clonal HSPC expansion,[72] whereas *NPM1* and *FLT3* mutations are more definitive events toward leukemogenic progression.[73] As fully transformed AML patient samples consist of multiple clones with distinct mutation loads, preleukemic mouse models can make unique contribution to disentangling the individual mutational effects despite their obvious drawback of not being human. Furthermore, future studies on other recurrent drivers of human preleukemia, such as *ASXL1*, would be important for a more comprehensive understanding of preleukemic hematopoiesis. In this growing field of preleukemic biology, our study establishes a novel analysis framework for integrated analysis of comprehensive single-cell genomics datasets and provides new lessons and opportunities for translation into individualized therapy for patients with AML.

### Limitations of the study

Although this study provides a powerful computational framework and data resources to understand preleukemic perturbations, we note a few limitations. First, this study's primary focus on LK HSPCs precluded an assessment of mutational impacts upon more differentiated cell populations. For instance, the known effects of *Ezh2* KO on B and T cell differentiation[25] are not captured within the confines of the LK gate. When focusing on mature cell populations, a more comprehensive reference atlas (e.g., a total mononuclear cell atlas) is required for robust data projection and subsequent downstream analyses. Second, although a broad spectrum of leukemogenic mutations has been characterized in this study, there still remains additional recurrent drivers of human preleukemia to be characterized (e.g., *ASXL1* and *TP53*).[47–49] The cooperative mechanisms of multiple mutations (e.g., combined *DNMT3A*, *NPM1*, and *FLT3* mutations) also need to be investigated to better understand the mechanisms of progression to overt leukemia. Finally, the insights gained from mouse models need to be compared with human preleukemia. To this end, recently developed methods enabling simultaneous single-cell DNA and RNA sequencing[37,74] represent promising approaches to extracting mutant-cell-specific features within preleukemic human donors.

### STAR★METHODS

Detailed methods are provided in the online version of this paper and include the following:

- KEY RESOURCES TABLE
- RESOURCE AVAILABILITY
  - Lead contact
  - Materials availability
  - Data and code availability
- EXPERIMENTAL MODEL AND STUDY PARTICIPANT DETAILS
  - Mice
- METHOD DETAILS

## Cell Genomics
### Resource

## SUPPLEMENTAL INFORMATION

## ACKNOWLEDGMENTS

Work in the Göttgens laboratory is funded by grants from Wellcome (206328/Z/17/Z), Blood Cancer UK (18002), Cancer Research UK (C1163/A21762), National Institutes of Health (NIDDK DK106766); core support grants from the Cancer Research UK Cambridge Centre (C49940/A25117), the Wellcome Trust (203151/Z/16/Z), and the UKRI Medical Research Council (MC_PC_17230). T.I. is supported by the Funai Foundation for Information Technology. Work in the Green laboratory is funded by grants from Wellcome (RG74909), Cancer Research UK (RG83389), and the William B. Harrison Foundation (RG91681). The authors thank Reiner Schulte, Chiara Cossetti, and Gabriela Grondys-Kotarba from the Cambridge Institute for Medical Research Flow Cytometry Core facility for their assistance with cell sorting. We would also like to thank the Cancer Research UK Cambridge Institute Genomics Core Facility for performing high-throughput sequencing. The graphical abstract was created using BioRender.com. For the purpose of open access, the author has applied a CC BY public copyright license to any author accepted manuscript version arising from this submission.

## AUTHOR CONTRIBUTIONS

Conceptualization, T.I., N.K.W., and B.G.; methodology, T.I., I.K., N.K.W., and B.G.; software, T.I., I.K., M.B., and X.W.; validation, T.I., N.K.W., and B.G.; formal analysis, T.I., N.K.W., and B.G.; investigation, T.I., S.C., M.S.V., K.H.S., M.J.W., and N.K.W.; resources, H.P.B., G.G., L.M., J.L., J.R., M.G., D.P., M.S.S., S.W., A.R.G., D.G.K., G.S.V., B.J.P.H., N.K.W., and B.G.; data curation, T.I., R.H., N.K.W., and B.G.; writing – original draft, T.I.; writing – review & editing, T.I., N.K.W., and B.G.; visualization, T.I., N.K.W., and B.G.; supervision, N.K.W. and B.G.; project administration, T.I., N.K.W., and B.G.; funding acquisition, T.I. and B.G.

## DECLARATION OF INTERESTS

Aspects of this work are included in United Kingdom patent application 2312684.0.

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

## STAR★METHODS

### KEY RESOURCES TABLE

| REAGENT or RESOURCE | SOURCE | IDENTIFIER |
|---|---|---|
| **Antibodies** | | |
| Anti-EPCR | STEMCELL technologies | 60038PE |
| Anti-CD48 | eBioscience | 17-0481-82 |
| Anti-CD150 | Biolegend | 115914 |
| Anti-*c*-Kit | Biolegend | 105826 |
| Anti-Sca1 | Biolegend | 108128 |
| Anti-CD45 | Biolegend | 103108 |
| Lineage Cocktail | STEMCELL technologies | 19856C (part of 19856) |
| Streptavidin | Biolegend | 405233 |
| 7AAD | Life technologies | A1310 |
| **Chemicals, peptides, and recombinant proteins** | | |
| Ammonium chloride | STEMCELL technologies | 07850 |
| EasySep™ Mouse Hematopoietic Progenitor Cell Isolation Kit | STEMCELL technologies | 19856 |
| **Critical commercial assays** | | |
| Chromium Single Cell 3′ Reagent Kit v2 | 10x Genomics | N/A |
| **Deposited data** | | |
| scRNA-seq data | This paper | GEO: GSE227026 |
| **Experimental models: Organisms/strains** | | |
| Mouse: Jak2 V617F: C57BL6J-*Jak2*$^{V617F/V617F}$ | Li et al.[12] | N/A |
| Mouse: Jak2 WT: C57BL6J-*Jak2*$^{+/+}$ | Li et al.[12] | N/A |
| Mouse: Calr 52-bp del: C57BL6J-*Calr*$^{fl/fl}$; *Vav-Cre*$^{+/−}$ | Li et al.[13] | N/A |
| Mouse: Calr WT: C57BL6J-*Calr*$^{fl/+}$; *Vav-Cre*$^{WT}$ | Li et al.[13] | N/A |
| Mouse: Calr 52-bp del: C57BL6J-*Calr*$^{fl/fl}$; *Mx1-Cre*$^{+/−}$ | Li et al.[13] | N/A |
| Mouse: Calr WT: C57BL6J-*Calr*$^{+/+}$; *Mx1-Cre*$^{+/−}$ | Li et al.[13] | N/A |
| Mouse: Flt3 ITD: C57BL6-Flt3$^{+/ITD}$ | Dovey et al.[14] | N/A |
| Mouse: Flt3 WT #1: C57BL6-Flt3$^{+/+}$ | Dovey et al.[14] | N/A |
| Mouse: Flt3 WT #2: C57BL6-CRTKO$^{WT}$; *Mx1-Cre*$^{WT}$ | Mesaeli et al.[75] | N/A |
| Mouse: Flt3 WT #3: C57BL6J-*Calr*$^{fl/+}$; *Vav-Cre*$^{WT}$ | Li et al.[13] | N/A |
| Mouse: Npm1c: C57BL6-*Npm1*$^{flox-cA/+}$; *Mx1-Cre*$^{+/−}$ | Vassiliou et al.[15] | N/A |
| Mouse: Npm1 WT: C57BL6- *Npm1*$^{flox-cA/+}$; *Mx1-Cre*$^{WT}$ | Vassiliou et al.[15] | N/A |
| Mouse: Idh1 R132H: C57BL6-*Idh1*$^{R132H/+}$; *Mx1-Cre*$^{+/−}$ | Gupta et al.[16] | N/A |
| Mouse: Idh1 WT: C57BL6-*Idh1*$^{R132H/+}$; *Mx1-Cre*$^{-}$ | Gupta et al.[16] | N/A |
| Mouse: Dnmt3a R882H: C57BL6-*Dnmt3a*$^{flox-R882H/+}$; *Mx1-Cre*$^{+/−}$ | Gozdecka et al.[17] | N/A |

*(Continued on next page)*

*Continued*

| REAGENT or RESOURCE | SOURCE | IDENTIFIER |
|---|---|---|
| Mouse: Dnmt3a WT: C57BL6-*Dnmt3a*<sup>flox-R882H/+</sup>; *Mx1-Cre*<sup>WT</sup> | Gozdecka et al.[17] | N/A |
| Mouse: Ezh2 KO: C57BL6-*Ezh2*<sup>fl/fl</sup>; *Mx1-Cre*<sup>+/−</sup> | Basheer et al.[18] | N/A |
| Mouse: Ezh2 WT: C57BL6-*Ezh2*<sup>fl/fl</sup>; *Mx1-Cre*<sup>WT</sup> | Basheer et al.[18] | N/A |
| Mouse: Utx KO: C57BL6-*Utx*<sup>fl/fl</sup>; *Mx1-Cre*<sup>+/−</sup> | Gozdecka et al.[19] | N/A |
| Mouse: Utx WT: C57BL6-*Utx*<sup>fl/fl</sup>; *Mx1-Cre*<sup>WT</sup> | Gozdecka et al.[19] | N/A |
| Software and algorithms | | |
| Code and algorithms for analysis | This paper | https://doi.org/10.5281/zenodo.8345465 |
| Cell Ranger | 10x Genomics | v6.0.1 |
| Scanpy | Wolf et al.[76] | https://scanpy.readthedocs.io/ |
| Scrublet | Wolock et al.[77] | https://github.com/swolock/scrublet |
| Seurat | Stuart et al.[20] | https://satijalab.org/seurat/ |
| MELD | Burkhardt et al.[21] | https://github.com/KrishnaswamyLab/MELD |
| CellRank | Lange et al.[9] | https://cellrank.readthedocs.io/ |
| scFEA | Alghamdi et al.[10] | https://github.com/changwn/scFEA |
| edgeR | Robinson et al.[78] | https://bioconductor.org/packages/release/bioc/html/edgeR.html |
| DESeq2 | Love et al.[79] | https://bioconductor.org/packages/release/bioc/html/DESeq2.html |
| tradeSeq | Van den Berge et al.[41] | https://github.com/statOmics/tradeSeq |
| GSEA | Subramanian et al.[80] | https://www.gsea-msigdb.org/gsea/index.jsp |

## RESOURCE AVAILABILITY

### Lead contact
Further information and requests for resources and reagents should be directed to and will be fulfilled by the lead contact, Berthold Göttgens (bg200@cam.ac.uk).

### Materials availability
This study did not generate new unique reagents.

### Data and code availability
- All raw sequencing data has been deposited on GEO under accession number GSE227026.
- All processed data and analysis results can be explored via our interactive web portal at https://gottgens-lab.stemcells.cam.ac.uk/preleukemia_atlas/.
- All original code has been deposited at https://github.com/TomoyaIsobe/PMCA/. DOI is listed in the key resources table.

## EXPERIMENTAL MODEL AND STUDY PARTICIPANT DETAILS

### Mice
This study included a total of 38 animals from eight different mutant mouse models: homozygous *Jak2* V617F[12] (n = 3), homozygous *Calr* 52-bp deletion[13] (n = 2), heterozygous *Flt3* ITD[14] (n = 3), heterozygous *Npm1c*[15] (n = 2), heterozygous *Idh1* R132H[16] (n = 3), heterozygous *Dnmt3a* R882H[17] (n = 2), homozygous *Ezh2* KO[18] (n = 2) and homozygous *Utx* KO[19] (n = 2). For each mutant model, paired wild-type mice with the same background (n = 1–3 per model; n = 17 in total) were simultaneously collected and subjected to flow cytometry sorting and sequencing to obtain wild-type comparators with minimal batch effects. For the *Flt3* ITD model, one paired and two unpaired wild-type animals were collected.[13,14,75] For the *Npm1*, *Idh1*, *Dnmt3a*, *Ezh2*, *Utx* and *Calr* (41-week-old mice) models,

*Mx1-Cre* was induced by intraperitoneal injection of pIpC (Sigma #P1530). All mice were bred and maintained in microisolator cages and provided continuously with sterile food, water, and bedding. All mice were kept in specified pathogen-free conditions, and all procedures were performed according to the United Kingdom Home Office regulations. Details of all animals used in this study, including age, sex and pIpC regimens, are summarized in Table S1. Further details on the individual mouse models are provided in the original reports cited above.

## METHOD DETAILS

### Flow cytometry sorting of hematopoietic stem and progenitor cells

Mouse bone marrow cells were collected from the femurs, tibiae and iliac crest and depleted of red blood cells by an ammonium chloride lysis step (STEMCELL Technologies). Cells were lineage depleted using EasySep Mouse Hematopoietic Progenitor Cell Isolation Kit (19856, STEMCELL Technologies). Lineage⁻ c-Kit⁺ (LK) cells were then isolated using the following antibodies (clone and company): streptavidin BV510 (BioLegend), c-kit APC-Cy7 (2B8, BioLegend), Sca1 BV421 (D7, BioLegend), CD45 FITC (30-F11, BioLegend), EPCR (CD201) PE (RMEPCR1560, STEMCELL Tech), CD150 PE/Cy7 (TC15-12F12.2, BioLegend), CD45 FITC (30-F1,1 BD Bioscience) and CD48 APC (HM48-1, eBioscience). Flow cytometry was performed on an LSRII Fortessa (BD) and all data were analyzed using FlowJo (BD). A representative gating strategy is shown in Figure S12.

### scRNA-seq data generation and preprocessing

The scRNA-seq data of the Calr mutant (n = 2) and the paired wild-type samples (n = 2) were previously published[27] and re-analyzed in this study; all remaining samples were newly sequenced for the current study. Single-cell sequencing libraries were generated using 10x Chromium (10x Genomics, Pleasanton, CA) reagent kit v2 according to the manufacturer's protocol and sequenced on an Illumina HiSeq 4000 or Illumina Novaseq 6000 platform. Raw reads were mapped to the mm10 genome and quantified using the Cell Ranger pipeline (v6.0.1) with default parameters. Cell-associated barcodes and background-associated barcodes were determined using the EmptyDrops method[81] implemented in the Cell Ranger pipeline, and the background-associated barcodes were excluded. Subsequent data analysis was performed using Scanpy.[76] Cell libraries with less than 1,000 detected genes or with mitochondrial gene expression exceeding 10% of UMI counts were removed from downstream analysis. Multiplets were estimated using the Python package Scrublet[77] and subsequently removed. Cell cycle phase was assigned to each cell using a previously published list of cell cycle-associated genes[82] and the Scanpy function 'tl.score_genes_cell_cycle'. All remaining cells that passed these quality controls (n = 269,048) were used for the downstream analysis. A previously published scRNA-seq dataset from *Tet2* KO and wild-type mice (GSE124822)[42] was subjected to the same mapping and preprocessing methods as shown above.

### Reference atlas of mouse hematopoietic landscape

Our previously published mouse HSPC atlas[11] including 44,802 LK and LSK cells was used as the reference hematopoietic landscape. The reference atlas data was log-normalized, and 5,000 highly variable genes were identified using the Scanpy function 'pp.highly_variable_genes'. Cell cycle-associated genes[83] were then removed and the expression values of the remaining 4,713 highly variable genes were scaled and used to compute 50 principal components, which were subsequently used to identify 10 nearest neighbors. The UMAP embedding was computed using the python package umap-learn[84] and the fitted model was saved to be applied to the mutant datasets as described below.

### Reference-based integration of mutant hematopoietic landscapes

All mutant and paired wild-type samples (n = 38) were log-normalized and projected onto the reference atlas using the reference-based integration method of Seurat.[20] First, all detected genes excluding cell cycle-associated genes were used to compute 30 canonical correlation vectors using the Seurat function 'RunCCA'. Next, integration anchors were identified using the 'FindIntegrationAnchors' with the option 'reference' set to the reference atlas data and using canonical correlation analysis for dimension reduction (reduction = 'cca'). Subsequently, each sample was integrated with the reference atlas using the 'IntegrateData' function using the canonical correlation vectors for anchor weighting. These integration steps were performed on the basis of the individual samples.

Integrated expression data were then scaled using the gene-wise means and standard deviations derived from the reference atlas data. Finally, 50 principal components and UMAP embeddings were computed using the same model fitted to the reference atlas data to project each sample onto the latent space of the reference atlas. Cell type annotation was transferred from the reference atlas using the Seurat functions 'FindTransferAnchors' and 'TransferData' with the dimension reduction method set to 'cca'.

### Differential abundance analysis

Mutation-specific changes in subpopulation abundance were quantified using the python package MELD.[21] First, a cell similarity graph was constructed based on the Euclidean distances in the common 50-dimensional principal component space after the data integration. Next, a kernel density estimate (KDE) was computed for each biological replicate and smoothed over the cell similarity graph with eight nearest neighbors and the smoothing parameter β = 15. The KDEs of paired mutant and wild-type samples

were then compared, and differential abundance was quantified as the relative likelihood of observing each cell in either the mutant or wild-type condition by performing cell-wise L1 normalization of the KDEs as implemented in the MELD package. The mutant relative likelihoods from all pairwise comparisons were averaged to obtain the mutation-specific differential abundance landscapes. Statistical significance was tested by comparing the sample-wise KDEs between the paired mutant and wild-type samples. We used paired t test for the mutant models with paired samples from multiple experimental batches (*Jak2*, *Calr*) and independent t test for the other datasets. p values were adjusted with the Benjamini-Hochberg (BH) procedure and cells with raw p values < 0.05 and BH-adjusted p values < 0.25 were considered significant.

### Differential fate probability analysis

Cellular fate probability was inferred using the python package CellRank.[9] The HSC score (molecular signature of long-term repopulating HSCs[22]) was first computed for all samples to determine the root cells for pseudotime calculation. The HSC score was then smoothed by taking the mean of 10 nearest neighbors in the diffusion map, and the cell with the highest smoothed HSC score was used as the root cell in each sample. Diffusion pseudotime was computed for each sample using the Scanpy function 'tl.dpt'. Cells without assigned cell types (cell type = 'Unassigned') were removed from the analysis.

Cell-to-cell transition probabilities were then inferred using the CellRank's pseudotime kernel ('tl.kernels.PseudotimeKernel') and the function 'compute_transition_matrix' with the 'soft' weighting scheme (threshold_scheme = 'soft'). Terminally differentiated cells were first identified in the reference atlas using the 'compute_macrostates' function. The terminal cells in each sample were then identified as the 10 nearest neighbors of the reference terminal cells in the common principal component space after the data integration. Estimated terminal cells with only one supporting neighbor or outlier terminal cells were excluded.

The fate probabilities toward these terminal states were computed using the CellRank's Generalized Perron Cluster Cluster Analysis (GPCCA) estimator and the function 'compute_absorption_probabilities'. The inferred fate probabilities were compared between the paired mutant and wild-type samples by logistic regression and likelihood ratio test using the Seurat function 'findMarkers' with the option 'LR' (test.use = 'LR'). The experimental batch information was included as a covariate in the logistic regression model to account for batch effects in the statistical evaluation. Lineage fate changes with |median fate probability differences| >5% and BH-adjusted p values < 0.05 were considered significant. The fate probabilities in each condition were summarized in pie charts with partition-based graph abstraction (PAGA) connections.[85]

### Differential metabolic flux analysis

Raw expression counts were first normalized to counts per million (CPM), which were then used as input for the python package scFEA.[10] Metabolic flux values for the 168 core metabolic reactions implemented in scFEA were inferred using the default parameters. The inferred metabolic flux values were compared between the paired mutant and wild-type samples by logistic regression and likelihood ratio test using the Seurat function 'findMarkers' with the option 'LR' (test.use = 'LR') with the experimental batch information included as a covariate. Constant low-flux reactions with differences between the minimum and maximum flux $<1.0 \times 10^{-4}$ were excluded. The size of differences between two groups was evaluated by Cohen's D standardized mean differences. The metabolic reactions with |Cohen's D| > 0.15 and BH-adjusted p values < 0.05 were considered significant. A published scRNA-seq dataset (GSE155763)[32] was used to validate the method.

### Pseudobulk differential expression analysis

Cell-type-wise differential expression analysis was performed with a pseudobulk method. First, gene-by-cell expression matrices were summed and aggregated for different cell types in different biological replicates to generate gene-by-replicate expression matrices. Single-cell-level mean expression was computed for each cell type, and low-expression genes with mean single-cell expression ≤0.05 counts per 10,000 UMIs (CP10K) were removed before differential expression testing. Differential expression analysis was performed using the likelihood ratio test of edgeR[78] with experimental batch information included as a covariate in the additive linear model. Genes with |log2 fold changes| ≥0.5 and BH-adjusted p values < 0.05 were considered significant. Gene set enrichment analysis was performed using the edgeR-derived −log10 adjusted p values with the signs of log2 fold changes as input for the preranked mode of GSEA software.[80]

### Pseudotemporal differential expression analysis

To identify genes with altered expression patterns over the course of differentiation, paired mutant and wild-type samples were combined and a common pseudotime was computed for each pair. The root cell for pseudotime inference was determined according to the highest smoothed HSC score as described above (see 'Differential fate probability analysis').

The erythroid, megakaryocytic and myelomonocytic trajectories were extracted from our Jak2, Calr and Flt3 datasets, respectively. The *Jak2* mutant or wild-type erythroid trajectory was defined as the cells belonging to the 'HSC', 'Immature prog', 'MEP', 'Early Ery prog', 'Mid Ery prog' or 'Late Ery prog' clusters and with the erythroid fate probability ≥0.20, neutrophil probability <0.3 and Mono/DC probability <0.23. A small number of cells (56 of 24,210 cells, 0.23%) at the end of the trajectory with variable pseudotime ≥0.40 were excluded. For the *Calr* mutant or wild-type megakaryocytic trajectory, cells belonging to the 'HSC', 'Immature prog', 'MEP' or 'MK prog' clusters and with the megakaryocytic fate probability ≥0.08 and neutrophil probability <0.3 were

included. The *Flt3* mutant or wild-type myelomonocytic trajectory was defined as the cells belonging to the 'HSC', 'Immature prog' or 'Mono/DC prog' clusters and with the monocytic fate probability ≥0.25 and the pseudotime <0.10.

Low-expression genes with mean CP10K ≤ 0.05 were removed before differential expression testing. A negative binomial generalized additive model was fitted using the 'fitGAM' function of the R package tradeSeq[41] with six knots (nknots = 6) and with the experimental batch information included as a covariate in the model. The fitted pseudotemporal expression patterns were then compared between the mutant and wild-type samples using the function 'conditionTest' with the log2 fold change threshold of 0.5 ('l2fc = 0.5'). Genes with BH-adjusted p values < 0.05 were considered significant.

### *CALR*-mutated patient dataset

A previously published scRNA-seq dataset (GSE117826)[37] of CD34$^+$ bone marrow HSPCs from patients with *CALR*-mutated essential thrombocythemia was analyzed. Multiplets were first estimated using Scrublet and subsequently removed. Cells with less than 1000 UMI counts or with mitochondrial gene percentage >10% were then removed. Cells with no genotyping UMI information were further removed. The remaining cells (n = 17,975) were log-normalized, and 500 highly variable genes were identified using the Scanpy function 'pp.highly_variable_genes'. Cell cycle-associated genes[86] were then removed and the expression values of the remaining 489 highly variable genes were scaled and used to compute 50 principal components. Five patient samples (ET01-ET05) were then integrated using Harmony[87] with a theta of 0. Harmony adjusted principal components were subsequently used to identify 10 nearest neighbors, and the UMAP embedding was computed using the Scanpy function 'tl.umap'. Leiden clustering was performed using the 'tl.leiden' function with a resolution of 0.6, and the cluster cell identity was assigned based on known marker genes. Based on the genotyping information obtained from GSE117826, cells with num.MUT.call ≥ 1 and total genotyping UMI ≥ 2 were labeled as mutant, while cells with num.MUT.call = 0 and total genotyping UMI ≥ 2 were labeled as wild-type.

For pseudotemporal differential expression analysis, the human HSC score[88] was first computed to determine the root cell for pseudotime calculation. The HSC score was then smoothed by taking the mean of 10 nearest neighbors in the diffusion map, and the cell with the highest smoothed HSC score was used as the root cell. Diffusion pseudotime was computed using the Scanpy function 'tl.dpt'. The megakaryocyte trajectory was defined as the HSC and megakaryocyte progenitor clusters. Cells at the end of the trajectory with variable pseudotime ≥0.10 were excluded, and the remaining cells (1,199 mutant and 2,226 wild-type cells) were analyzed for pseudotemporal differential gene expression using tradeSeq.

### AML patient datasets

Publicly available TCGA AML,[1] Beat AML,[2] EGAD00001008484[55] and TARGET AML[56] RNA-seq datasets were used. Gene expression raw counts and TPM values of the TCGA and Beat AML cohorts were downloaded using the R package TCGAbiolinks. Clinical data for the TCGA cohort were downloaded from the National Cancer Institute Genomic Data Commons (https://gdc.cancer.gov/about-data/publications/laml_2012). Clinical and mutation data for the Beat AML cohort were downloaded from the Beat AML data portal (https://biodev.github.io/BeatAML2). For the Beat AML cohort, *de novo* AML patients with RNA-seq data and FAB classification available were included. Samples with low tumor cell content (<20%) were excluded. Patients with FAB labels of "M0/M1", "M4eo", "M5a" and "M5b" were reannotated in the figures as "M1", "M4", "M5" and "M5", respectively. For the EGAD00001008484 dataset, gene expression data were downloaded from the European Genome-Phenome Archive (EGA), and *de novo* AML patients were included. For the TARGET AML cohort, gene expression data were downloaded from the National Cancer Institute TARGET data portal (https://target-data.nci.nih.gov/Public/AML/).

### Analysis of patient RNA-seq data

The gene expression matrices were filtered to include only the 216 preleukemic lineage perturbation signature genes, and low-expression genes with mean TPM ≤0.5 were further excluded. The expression TPM values were log-normalized, scaled and used to compute the principal components. The first three principal components were used for visualization. For the TCGA cohort, hierarchical clustering was performed using the 'ward.D2' method and differential expression analysis were performed using the R package DESeq2.[79] Two-group comparisons were performed using the Wald test, and *p* values were adjusted with the BH procedure. The Stem11 genes were defined based on significant overexpression (log2 fold change > 2 and BH-adjusted p < 0.05) in the Stem cluster of the TCGA cohort, where 12 genes were identified and one gene (*RORB*) was removed due to low expression in the Beat-AML cohort (mean TPM <0.5). The Stem11 score was calculated as the sum of the Z-scores of the Stem11 genes, and Stem-11 high patients were defined as the top 15th percentile.

### Survival analysis

Survival was estimated using the Kaplan–Meier method and the difference was tested using the log rank test. Overall survival was compared over the first 6 years of observation. For multivariate analysis, a Cox proportional hazards regression model was used to identify the risk factors associated with the overall survival. p values of less than 0.05 were considered statistically significant. Statistical analysis was performed using the R package survival.

 **CellPress**

**Cell Genomics**
**Resource**

## QUANTIFICATION AND STATISTICAL ANALYSIS

Statistical analyses were performed using Python (3.8.6 or 3.8.12) or R (3.6.3 or 4.0.3). All statistical methods used in this study are described in the individual figure legends and the METHOD DETAILS. For differential abundance analysis, cells with raw p values < 0.05 and BH-adjusted p values < 0.25 were considered significant. For differential fate probability analysis, differential metabolic flux analysis and differential expression analysis, BH-adjusted p values of less than 0.05 were considered significant. For all the other analyses, raw p values of less than 0.05 were considered significant.

