## [Document S2. Transparent peer review records for Isobe et al. · Cell Genomics]

Preleukemic hematopoietic stem/progenitor single-cell-landscapes reveal mutationspecific mechanisms and preleukemic programs predictive of AML patient outcomes

Tomoya Isobe, Iwo Kucinski, Melania Barile, Xiaonan Wang, Rebecca Hannah, Hugo P. Bastos, Shirom Chabra, M. S. Vijayabaskar, Katherine Hm Sturgess, Matthew J. Williams, George Giotopoulos, Ludovica Marando, Juan Li, Justyna Rak, Malgorzata Gozdecka, Daniel Prins, Mairi S. Shepherd, Sam Watcham, Anthony R. Green, David G. Kent, George S. Vassiliou, Brian J. P. Huntly, Nicola K. Wilson, Berthold Göttgens

Summary

Initial submission: Received : Apr 14 2023

Scientific editor: Sara Rohban

First round of review: Number of reviewers: 2
Revision invited : May 15th , 2023
Revision received : Jul 13th , 2023

Second round of review: Number of reviewers: 3
Revision invited : Aug 16th , 2023
Revision received : Sep 01st, 2023

Third round of review: Number of reviewers: 1
Accepted : Sep 29th , 2023

Data freely available: YES

Code freely available: YES

This transparent peer review record is not systematically proofread, type-set, or edited. Special characters, formatting, and equations may fail to render properly. Standard procedural text within the editor's letters has been deleted for the sake of brevity, but all official correspondence specific to the manuscript has been preserved.

Referees' reports, first round of review

Reviewer #1: This manuscript presents an analysis of single cell RNA-seq data of a variety of mouse models of mutations seen in myeloid neoplasms. The data were generated from lineage negative cells prior to overt disease. The authors draw a variety of conclusions and generate a gene signature which they claim is predictive of outcome in patients with acute myeloid leukemia. Overall, it is not clear what solid conclusions are being made from this study as many of the points are not substantiated by data other than gene expression analysis.

-The Abstract is opaque and does not clearly explain that this was a study of mouse models of myeloid neoplasms. The statement that "how individual mutations perturb the entire hematopoietic system towards malignant phenotypes remains elusive" is quite arguable as there is quite a large body of work on how, as one example, mutations in DNMT3A or IDH1/2 perturb hematopoiesis and gene expression. A more accurate summary of the actual goals of this study would be much more helpful.

-It is not clear what evidence there is for the last sentence of the Introduction that this study "establishes a novel framework for translating preleukemic biology into personalized treatment." In what way do the results from the present manuscript provide insights into "personalized treatment"?

-Mutations which are the most abundant in CH including TET2 and ASXL1 mutations were not included and conversely, mutations which are rarely seen in clonal hematopoiesis (FLT3, NPM1, EZH2, and UTX mutations) were included.

-The metabolic data in Figure 4B-E would need to be substantiated by actual evidence of altered metabolite usage. The current conclusions on metabolism are made based solely on gene expression data and the claims that metabolic changes may result in impaired differentiation appear to be hypotheses and not substantiated by data. Moreover, the figure legend erroneously describes these data as depicting "glycolysis and TCA activities" when it appears to be gene expression of components of these pathways which are depicted.

-Do the gene expression characteristics of the individual mutant mouse models relate to the characteristics of patient samples with the same mutations?

Reviewer #2: Comments enter in this field will be shared with the author; your identity will remain anonymous.

Summary

How pre-leukemic mutations alter the complex phenotype of immature bone marrow cells is currently poorly understood. The authors studied this topic by generating nearly 255,000 single cell transcriptomes from hematopoietic stem- and progenitor cells obtained from 36 mice, carrying 8 distinct pre-leukemic mutations and their wild-type counterparts. State-of-the art data analysis techniques allowed the authors to visualize and extract mutation specific transcriptional changes, affecting e.g., central metabolism and cell fate transitions.

The authors defined a new "stemness" signature from their data that correlates with the most aggressive, immature AML-subtypes, with adverse prognosis, both in adult and pediatric AML. The manuscript and figures are accessible to both experts in the field and those with a general understanding of molecular biology and genetics. Taken together, this is an interesting and well-

designed study. I recommend publication of a revised manuscript that addressed the comments below.

Main comment

The differential abundance in the HSC (Fig. 2C) and neutrophil compartments (Fig. 2F) of the FLT3-ITD model are interesting, but comprise a relatively small number of cells, obtained from a comparison with a single WT replicate. To exclude the possibility that this is a technical or batch-related artifact, the authors should include at least another WT replicate.

Minor comments

Line 198 - 205: the authors suggest that the reason for downregulation of glycolysis and the TCA cycle in certain models (Idh1, Npm1, Ezh2 and Utx) could be related to epigenetic processes. While metabolic changes may be required to provide the necessary energy changes and/or substrates for epigenetic processes, neither metabolism (at best inferred), nor epigenetic processes have truly been measured here. Therefore, the (speculative) reasoning provided here may be best moved to the discussion section.

Line 265-267: What is the rationale for selecting lineage signatures from only the JAK2, CALR and FLT3-ITD? Other models also displayed strong cell fate bias. E.g., the UTX KO model revealed neutrophil bias (Fig. 2A) with many DEGs in the HSC and progenitor compartments (Fig. 5A).

Authors' response to the first round of review

We would like to thank the reviewers for their constructive comments, which helped us to further improve our manuscript. We have conducted additional data analysis and experiments and revised parts of our manuscript where appropriate. We have provided a point-by-point response below in blue font and marked all changes in the manuscript text in red font. The references we cite in our response are listed at the end of this letter.

Reviewer #1:

This manuscript presents an analysis of single cell RNA-seq data of a variety of mouse models of mutations seen in myeloid neoplasms. The data were generated from lineage negative cells prior to overt disease. The authors draw a variety of conclusions and generate a gene signature which they claim is predictive of outcome in patients with acute myeloid leukemia. Overall, it is not clear what solid conclusions are being made from this study as many of the points are not substantiated by data other than gene expression analysis.

- The Abstract is opaque and does not clearly explain that this was a study of mouse models of myeloid neoplasms. The statement that "how individual mutations perturb the entire hematopoietic system towards malignant phenotypes remains elusive" is quite arguable as there is quite a large body of work on how, as one example, mutations in DNMT3A or IDH1/2 perturb hematopoiesis and gene expression. A more accurate summary of the actual goals of this study would be much more helpful.

We thank the reviewer for raising this point. We agree that there exists a growing body of research on the mechanisms of mutation-driven perturbations, which was inadequately acknowledged in our original Abstract. We have revised the Abstract to explicitly delineate the existing knowledge gaps and the goals of our current study (lines 24-38).

- It is not clear what evidence there is for the last sentence of the Introduction that this study "establishes a novel framework for translating preleukemic biology into personalized treatment." In what way do the results from the present manuscript provide insights into "personalized treatment"?

We thank the reviewer for this comment; our single-cell transcriptomic analysis of preleukemic mouse models has defined the Stem11 signature, which possesses the potential to refine the prospective risk stratification strategy for AML patients (Figure 6I). In this regard, we argued for an improved treatment stratification guided by molecular understanding of preleukemic biology. However, we agree with the reviewer that the term "personalized treatment" may encompass a broader scope; therefore, we have revised this sentence to specifically represent our finding of the Stem11 signature and its potential for improved risk stratification of AML as follows:

"This single-cell-resolution multi-scale analysis illustrates mutation-specific mechanisms of hematopoietic perturbation and identifies preleukemic genetic programs predictive of AML patient outcomes (Stem11 signature), thus establishing a novel framework for translating preleukemic biology into an improved treatment stratification strategy for AML patients." (Lines 72-76)

- Mutations which are the most abundant in CH including TET2 and ASXL1 mutations were not included and conversely, mutations which are rarely seen in clonal hematopoiesis (FLT3, NPM1, EZH2, and UTX mutations) were included.

We thank the reviewer for this comment; as the reviewer points out, the current study has primarily focused on leukemogenic mutations recurrently found in myeloid neoplasms. Through this approach, we have successfully characterized the perturbation effects of individual mutations, which are often complicated by co-existing mutations in leukemic patient samples. Since our

methodology would be applicable to any mutant mouse models, we agree with the reviewer regarding the importance of investigating additional mutations, including those associated with clonal hematopoiesis, which we would like to pursue in a future study.

- The metabolic data in Figure 4B-E would need to be substantiated by actual evidence of altered metabolite usage. The current conclusions on metabolism are made based solely on gene expression data and the claims that metabolic changes may result in impaired differentiation appear to be hypotheses and not substantiated by data. Moreover, the figure legend erroneously describes these data as depicting "glycolysis and TCA activities" when it appears to be gene expression of components of these pathways which are depicted.

We thank the reviewer for raising this point. We also very much appreciate the comment about hypothesis generation, since this is very much the point of papers reporting comprehensive datasets such as ours. We would also like to note the editor's instruction to us that the revision should be returned within 2 months, thus precluding comprehensive experimental validation of hypotheses.

We nevertheless attempted to identify specific compounds which would inhibit phosphofructokinase 1 Platelet isoform (pfkp), as the gene for this enzyme was specifically upregulated in the metabolic flux analysis. Exposure of LT-HSCs cultured in vitro for 8 days with an inhibitor of this enzyme (Querticin) led to a dose-dependent reduction in absolute cell numbers. Moreover, the number of lin⁺ cells showed a more profound drop (and lin⁻ a less profound drop), consistent with the notion that the inhibitor slowed down differentiation. Given the effect on overall cell numbers however, we do not feel confident to disentangle a possible link between differentiation rate and reduction of glycolysis from overall toxicity of the compound (please see Reviewer Figure 1 below this answer).

To nevertheless address the reviewer's point about corroborating metabolic predictions, we returned to the transcriptomic datasets and compared the cell cycle phase associated with the increased cell abundance due to the Group 1 and Group 2 mutations. Intriguingly, despite both the *Calr* mutation and *Utx* KO increasing the abundance of megakaryocyte progenitors (Figure 2A), they exhibited contrasting effects on the cell cycle status. Specifically, the *Calr* mutation led to an increased proportion of cycling G₂/M-phase cells, whereas the *Utx* KO resulted in an increase in G₁-phase cells (please see new Figure S4F). In addition to contrasting effects on cell cycle, the two models also showed opposing predictions with regard to glycolysis, which is predicted to be up in *Calr* and down in *Utx*. Importantly, these contrasting predictions on glycolysis are perfectly in line with the predicted proliferation (higher proliferation = higher glycolysis and vice versa). This analysis therefore demonstrates how metabolic predictions are backed up by proliferation

scores, and thus sheds further light on the potential proliferative and repressive effects of the Group 1 and Group 2 mutations, respectively. We have now included these results in the manuscript text (lines 362-366).

Reviewer Figure 1: FACS analysis of wild-type ESLAM HSCs treated with Quercetin for 8 days in myeloid promoting culture. Total numbers of live cells and lineage positive cells were reduced in a dose dependent manner.

However, as we acknowledge the reviewer's point regarding the absence of direct metabolite measurements in our study, we have clarified in the figure that our results show "predicted" metabolic activities (please see updated Figure 4) and moved our speculation on the possible association between the predicted metabolic changes and differentiation to the Discussion section (lines 366-370).

- Do the gene expression characteristics of the individual mutant mouse models relate to the characteristics of patient samples with the same mutations?

We thank the reviewer for this comment, and we have now assessed the gene expression similarity between our mouse models and human patients. Firstly, we have shown that patient-derived gene signatures are significantly enriched in our corresponding mouse models (please see new Figure S5A), indicating that key gene expression signature of each disease subtype are recapitulated in the corresponding mouse model. Since the *Idh1* R132H model has only <10 differentially expressed genes in all cell types, gene set enrichment was not evaluable; however, this small transcriptomic effect is consistent with previous observations in expression profiling of AML patients^{1,2}, where *IDH1*-mutated patients do not form a transcriptionally distinct cluster but are very heterogenous. For the *Dnmt3a* R882H model, we evaluated the enrichment of

DNMT3A R882 mutation-associated signatures obtained from single-cell transcriptomic and genotyping analysis of human donors with clonal hematopoiesis³. Although we observed a trend towards the expected up- and down-regulation of these signatures (Reviewer Figure 2), the enrichment was not statistically significant. This may be due to the less pronounced hematopoietic perturbation by this mutation in comparison to the other leukemogenic mutations, as reflected by the second lowest number of differentially expressed genes (Figure 5A).

Reviewer Figure 2. Enrichment analysis of clonal hematopoiesis donor-derived *DNMT3A* signature in our model. NES, normalized enrichment score.

Furthermore, we have analyzed a previously published single-cell transcriptomic and genotyping dataset of CD34⁺ bone marrow HSPCs from patients with *CALR*-mutated essential thrombocythemia⁴. This analysis demonstrated consistent pseudotemporal gene expression patterns shared between our mouse model and human patients, particularly affecting the unfolded protein response genes, previously implicated in *CALR*-mutated myeloproliferative neoplasms⁴ (please see new Figure S5B–S5F).

Overall, we hope that these bulk and single-cell-level consistency with patient data provides additional validation of the clinical relevance of our mouse models. The original reports in which we established these mouse models^{5–12} provide further phenotypic validations. We have now included the methods (lines 584–608) and results (lines 218–220 and 257–261) of these additional analyses in the manuscript text.

Reviewer #2:

Summary

How pre-leukemic mutations alter the complex phenotype of immature bone marrow cells is currently poorly understood. The authors studied this topic by

generating nearly 255,000 single cell transcriptomes from hematopoietic stem- and progenitor cells obtained from 36 mice, carrying 8 distinct pre-leukemic mutations and their wild-type counterparts. State-of-the art data analysis techniques allowed the authors to visualize and extract mutation specific transcriptional changes, affecting e.g., central metabolism and cell fate transitions.

The authors defined a new "stemness" signature from their data that correlates with the most aggressive, immature AML-subtypes, with adverse prognosis, both in adult and pediatric AML. The manuscript and figures are accessible to both experts in the field and those with a general understanding of molecular biology and genetics. Taken together, this is an interesting and well designed study. I recommend publication of a revised manuscript that addressed the comments below.

Main comment

The differential abundance in the HSC (Fig. 2C) and neutrophil compartments (Fig. 2F) of the FLT3-ITD model are interesting, but comprise a relatively small number of cells, obtained from a comparison with a single WT replicate. To exclude the possibility that this is a technical or batch related artifact, the authors should include at least another WT replicate.

We thank the reviewer for this important suggestion, and we have now analyzed two additional wild-type animals to ensure the robustness of our differential abundance analysis for the Flt3 dataset. Consequently, our original finding of differential abundance in the HSCs and neutrophil progenitors has been successfully confirmed (please see updated Figure 2). We have updated the statistical results described in the Results section (lines 114-126). Furthermore, since the total number of animals and cells analyzed in this study has increased (now 269,048 cells from 38 animals), we have updated Figure 1 and S1 accordingly.

Minor comments

Line 198 - 205: the authors suggest that the reason for downregulation of glycolysis and the TCA cycle in certain models (Idh1, Npm1, Ezh2 and Utx) could be related to epigenetic processes. While metabolic changes may be required to provide the necessary energy changes and/or substrates for epigenetic processes, neither metabolism (at best inferred), nor epigenetic processes have truly been measured here. Therefore, the (speculative) reasoning provided here may be best moved to the discussion section.

We thank the reviewer for this comment, and we have now moved our interpretation of the predicted metabolic alterations to the Discussion section (lines 366-370).

Line 265-267: What is the rationale for selecting lineage signatures from only the JAK2, CALR and FLT3-ITD? Other models also displayed strong cell fate bias. E.g., the UTX KO model revealed neutrophil bias (Fig. 2A) with many DEGs in the HSC and progenitor compartments (Fig. 5A).

Thank you for raising this point. We have selected the *Jak2*, *Calr* and *Flt3* mutations since our metabolic analysis and the existing literature^{5,6,13} suggest their role as active drivers of lineage bias as opposed to inducing passive accumulation due to differentiation block. Consequently, our PLPS genes represent a comprehensive signature of biased differentiation towards the major myeloid differentiation trajectories (i.e., erythroid, megakaryocytic and myelomonocytic lineages).

To clarify this, we have added the following explanations to the text:

“Since our analysis and previous reports indicate that the *Jak2*, *Calr* and *Flt3* mutations actively drive proliferation and biased differentiation, we combined the erythroid, megakaryocytic and myelomonocytic bias signatures dysregulated by these mutations (Table S3). This allowed us to develop a comprehensive gene signature representing the biased differentiation towards the major myeloid lineages (hereafter, preleukemic lineage perturbation signature (PLPS); 216 genes).” (Lines 268-273)

According to the reviewer’s suggestion, we have also evaluated the performance of an updated PLPS signature that incorporates the DEGs in the *Utx* KO neutrophilic trajectory. Although this signature identified 4 clusters comparable to the original clustering (Reviewer Figure 3A), no significant survival differences were observed (Reviewer Figure 3B). Furthermore, this updated PLPS signature resulted in the segregation of acute promyelocytic leukemia (APL or AML M3) patients into two distinct clusters (Reviewer Figure 3A), despite these patients having a distinct global molecular profile compared to other AML subtypes¹. This may be due to the considerably higher number of DEGs in the *Utx* KO model compared to the other models (552 genes in the *Utx* model vs. 80, 94 and 68 genes in the *Jak2*, *Calr* and *Flt3* models, respectively), which made the updated PLPS genes more sensitive to granulocytic differentiation status. Considering the broader range of myeloid differentiation status observed in AML samples, we have concluded that our original PLPS signature provides a better representation of AML patient data in terms of myeloid differentiation status and prognosis.

Reviewer Figure 3. (A) Hierarchical clustering of the TCGA AML cohort using the updated PLPS genes including the DEGs in the *Utx* KO neutrophilic trajectory. (B) Survival analysis comparing the updated clusters.

References

1. The Cancer Genome Atlas Research Network. Genomic and Epigenomic Landscapes of Adult De Novo Acute Myeloid Leukemia. *New Engl J Medicine* 368, 2059–2074 (2013).
2. Cheng, W.-Y. *et al.* Transcriptome-based molecular subtypes and differentiation hierarchies improve the classification framework of acute myeloid leukemia. *Proc. Natl. Acad. Sci.* 119, e2211429119 (2022).
3. Nam, A. S. *et al.* Single-cell multi-omics of human clonal hematopoiesis reveals that DNMT3A R882 mutations perturb early progenitor states through selective hypomethylation. *Nat. Genet.* 54, 1514–1526 (2022).
4. Nam, A. S. *et al.* Somatic mutations and cell identity linked by Genotyping of Transcriptomes. *Nature* 571, 355–360 (2019).
5. Li, J. *et al.* JAK2V617F homozygosity drives a phenotypic switch in myeloproliferative neoplasms, but is insufficient to sustain disease. *Blood* 123, 3139–3151 (2014).
6. Li, J. *et al.* Mutant calreticulin knockin mice develop thrombocytosis and myelofibrosis without a stem cell self-renewal advantage. *Blood* 131, 649–661 (2018).
7. Dovey, O. M. *et al.* Identification of a germline F692L drug resistance variant in cis with *Flt3*-internal tandem duplication in knock-in mice. *Haematologica* 101, e328–e331 (2016).

8. Vassiliou, G. S. *et al.* Mutant nucleophosmin and cooperating pathways drive leukemia initiation and progression in mice. *Nat Genet* 43, 470–475 (2011).
 9. Gupta, S. *et al.* Transcriptional variability accelerates preleukemia by cell diversification and perturbation of protein synthesis. *Sci Adv* 8, eabn4886 (2022).
 10. Gozdecka, M. *et al.* Genetic Vulnerabilities of DNMT3AR882H in Myeloid Malignancies. *Blood* 134, 111–111 (2019).
 11. Basheer, F. *et al.* Contrasting requirements during disease evolution identify EZH2 as a therapeutic target in AML. *J Exp Medicine* 216, 966–981 (2019).
 12. Gozdecka, M. *et al.* UTX-mediated enhancer and chromatin remodeling suppresses myeloid leukemogenesis through noncatalytic inverse regulation of ETS and GATA programs. *Nature Genetics* 50, 1–27 (2018).
 13. Lee, B. H. *et al.* FLT3 Mutations Confer Enhanced Proliferation and Survival Properties to Multipotent Progenitors in a Murine Model of Chronic Myelomonocytic Leukemia. *Cancer Cell* 12, 367–380 (2007).
-

Referees' report, second round of review

Reviewer #1: The authors have addressed my initial comments and concerns. I have no further issues with the manuscript.

Reviewer #2: The authors have addressed all our issues/suggestions in their revised manuscript.

Reviewer #3: The title of this manuscript is "Preleukemic hematopoietic stem/progenitor single cell landscapes reveal mutation-specific mechanisms and preleukemic programs predictive of AML patient outcomes".

What this manuscript presents is scRNA seq analysis of 8 mouse models of perturbed hemopoiesis. The data is well analyzed. The data's main value is as a resource. It's value in understanding human preleukemia, and AML specifically, is more limited.

90% of human preleukemia (strictly myeloid preleukemia) arises from mutations in human stem cells in DNMT3A, TET2 and ASXL1 (here, I am defining preleukemia as a clonal condition that predisposes to AML). Mutations in NPM1 are seen in ~20-30% of AML patients but the preleukemic phase is rarely captured as progression to AML is often fast. IDH1 mutations can cause preleukemia (~2-5%) and are seen in ~ 5-10% of AML patients.

So, the main mouse models of interest are Dnmt3aR882H and Npm1c, IdhR132H. JAK2V617F and Calr52 bp deletion are common in myeloproliferative disorders (MPD), rather than preleukemia that leads to AML. These two mutations are seen in 5-10% of AML; and a specific subtype of AML that which is secondary to clinical MPD. The relevance of Utx KO and Ezh2 KO

to preleukemia and AML is more questionable.

There are two main issues the authors need to make clear:

1. The authors need to be much more careful in how they interpret the value of their data with respect to human preleukemia and AML. To give examples:
 - a. Stress the limitation of not having TET2 and ASXL1 mutant datasets.
 - b. Stress the limitation of Jak2 and Calr mutant datasets - they are relevant to AML secondary to MPD.
 - c. Stress the limitation of the Utx and Ezh2 Ko datasets.
 - d. At Reviewer 1's reasonable request, the authors compared their datasets to CALR mutant CH myelofibrotic patient (Ref 37; Nam et al Nature Medicine 2019), erythroid lineage cells from a JAK2V617F mutant MPD patients (Chen et al Cancer Cell 2010) and old bulk bone marrow mononuclear RNA dataset using Affymetrix U133A oligonucleotide arrays from AML patients (the data from Valk lab in 2004). These are not common preleukemia datasets (DNMT3A, TET2, AXL1 mutant preleukemia accounts for ~90% of human preleukemia). The AML datasets are not relevant to preleukemia as AML cells will have multiple fully transformed clones. So, there are obvious limitations of this comparison.
 2. The independent prognostic value of the Stem-11 signature is unclear. Multiple factors determine a patient's prognosis: age, performance status (PS), treatment received and genetic and molecular subtypes. To properly establish if Stem-11 has independent prognostic significance the authors should use RNA seq data from patients treated on large clinical trials (i.e. uniform group of patients with respect to age and performance status), where uniform treatment is given (e.g. intensive chemotherapy±allotransplant or HMA+ven treatment trials) and conduct both univariate and multivariate analysis, to establish if Stem-11 has independent prognostic value, over and above, age, PS, treatment, cytogenetic and molecular subtype (ELN 2022 would be a good surrogate for this).
-

Authors' response to the second round of review

We were delighted to read that reviewers #1 and #2 were satisfied with our revised manuscript. We would also like to thank reviewer #3 for their additional constructive comments, all of which have now been addressed as outlined in the specific point-by-point response below.

Reviewer #1: The authors have addressed my initial comments and concerns. I have no further issues with the manuscript.

Thank you very much for your previous suggestions which we think have improved the paper.

Reviewer #2: The authors have addressed all our issues/suggestions in their revised manuscript.

Thank you very much for your previous suggestions which we think have improved the paper.

Reviewer #3: The title of this manuscript is "Preleukemic hematopoietic stem/progenitor single cell landscapes reveal mutation-specific mechanisms and preleukemic programs predictive of AML patient outcomes".

What this manuscript presents is scRNA seq analysis of 8 mouse models of perturbed hemopoiesis. The data is well analyzed. The data's main value is as a resource. It's value in understanding human preleukemia, and AML specifically, is more limited.

90% of human preleukemia (strictly myeloid preleukemia) arises from mutations in human stem cells in DNMT3A, TET2 and ASXL1 (here, I am defining preleukemia as a clonal condition that predisposes to AML). Mutations in NPM1 are seen in ~20-30% of AML patients but the preleukemic phase is rarely captured as progression to AML is often fast. IDH1 mutations can cause preleukemia (~2-5%) and are seen in ~ 5-10% of AML patients.

So, the main mouse models of interest are Dnmt3aR882H and Npm1c, IdhR132H. JAK2V617F and Calr52 bp deletion are common in myeloproliferative disorders (MPD), rather than preleukemia that leads to AML. These two mutations are seen in 5-10% of AML; and a specific subtype of AML that which is secondary to clinical MPD. The relevance of Utx KO and Ezh2 KO to preleukemia and AML is more questionable.

There are two main issues the authors need to make clear:

1. The authors need to be much more careful in how they interpret the value of their data with respect to human preleukemia and AML. To give examples:
 - a. Stress the limitation of not having TET2 and ASXL1 mutant datasets.

We thank the reviewer for raising this point. We have now analyzed a previously published scRNA-seq dataset from a *Tet2* KO model¹ using our analysis pipeline. Reassuringly, differential abundance and fate probability analyses identified expansion of HSCs and myeloid-biased hematopoiesis within the *Tet2* KO model (please see new Figures S7A-S7C), consistent with previous observations²⁻⁴. Furthermore, differential metabolic flux analysis and gene expression analysis revealed reduced activity of energy generating metabolic pathways and altered expression patterns of myeloid differentiation/maturation markers in the *Tet2* KO HSPCs (new Figures S7D-S7G), thus adding to previously reported insights into the molecular scale alterations associated with impaired hematopoietic differentiation by the *Tet2*

mutation³⁻⁵. Furthermore, this analysis underscores the robustness of wider applicability of our analysis pipeline by highlighting the ease of extracting information from external datasets. We have now included these results in the manuscript text (lines 264-279).

Since there are no publicly available *ASXL1* mutant datasets, we have mentioned this point in the Discussion section as a limitation of our current study (lines 439-441).

b. Stress the limitation of *Jak2* and *Calr* mutant datasets - they are relevant to AML secondary to MPD.

We thank the reviewer for this comment; we have now stressed the specificity of *JAK2* and *CALR* mutations to MPD and secondary AML in the Discussion section (lines 433-434).

c. Stress the limitation of the *Utx* and *Ezh2* Ko datasets.

We thank the reviewer for this comment. We have now mentioned the rare occurrence of the *UTX* and *EZH2* mutations in the Discussion section (lines 434-435).

d. At Reviewer 1's reasonable request, the authors compared their datasets to *CALR* mutant CH myelofibrotic patient (Ref 37; Nam et al Nature Medicine 2019), erythroid lineage cells from a *JAK2V617F* mutant MPD patients (Chen et al Cancer Cell 2010) and old bulk bone marrow mononuclear RNA dataset using Affymetrix U133A oligonucleotide arrays from AML patients (the data from Valk lab in 2004). These are not common preleukemia datasets (*DNMT3A*, *TET2*, *AXL1* mutant preleukemia accounts for ~90% of human preleukemia). The AML datasets are not relevant to preleukemia as AML cells will have multiple fully transformed clones. So, there are obvious limitations of this comparison.

We thank the reviewer for raising this point. We agree with the reviewer that the fully transformed AML samples have multiple clones and mutations and AML datasets are therefore not perfect human counterparts to validate our preleukemic mouse models. Ideally, gene signatures derived from comparisons of mutant and wild-type cells in preleukemic human donors should be used, but this was not available for the current study. Therefore, according to the reviewer's comment, we have now mentioned this limitation in the manuscript text (lines 436-439).

2. The independent prognostic value of the Stem-11 signature is unclear. Multiple factors determine a patient's prognosis: age, performance status (PS),

treatment received and genetic and molecular subtypes. To properly establish if Stem-11 has independent prognostic significance the authors should use RNA seq data from patients treated on large clinical trials (i.e. uniform group of patients with respect to age and performance status), where uniform treatment is given (e.g. intensive chemotherapy±allotransplant or HMA+ven treatment trials) and conduct both univariate and multivariate analysis, to establish if Stem-11 has independent prognostic value, over and above, age, PS, treatment, cytogenetic and molecular subtype (ELN 2022 would be a good surrogate for this).

We thank the reviewer for this comment. Although the TCGA, Beat AML and TARGET datasets are valuable resources, it is true that they are not from single clinical trials with the same inclusion/exclusion criteria and uniform treatment regimens. According to the reviewer's suggestion and thanks to the recently published RNA-seq dataset of AML patients uniformly treated with intensive chemotherapy⁶, we have now evaluated the independent prognostic significance of the Stem11 signature. As a result, Stem11 classification showed a significant association with overall survival in both univariate ($P = 7.2 \times 10^{-3}$) and multivariate analysis ($P = 7.5 \times 10^{-3}$) including patient age and ELN2017 cytogenetic/molecular risk classification (please see new Figures S10I and S10J), although the performance status data were not available for this cohort. We have now included these results in the manuscript text (lines 336-340).

References

1. Izzo, F. *et al.* DNA methylation disruption reshapes the hematopoietic differentiation landscape. *Nat Genet* **52**, 378–387 (2020).
2. Ko, M. *et al.* Impaired hydroxylation of 5-methylcytosine in myeloid cancers with mutant TET2. *Nature* **468**, 839–843 (2010).
3. Moran-Crusio, K. *et al.* Tet2 Loss Leads to Increased Hematopoietic Stem Cell Self-Renewal and Myeloid Transformation. *Cancer Cell* **20**, 11–24 (2011).
4. Nakauchi, Y. *et al.* The Cell Type–Specific 5hmC Landscape and Dynamics of Healthy Human Hematopoiesis and TET2 -Mutant Preleukemia. *Blood Cancer Discov.* **3**, 346–367 (2022).
5. Encabo, H. H. *et al.* Loss of TET2 in human hematopoietic stem cells alters the development and function of neutrophils. *Cell Stem Cell* **30**, 781-799.e9 (2023).

Jayavelu, A. K. *et al.* The proteogenomic subtypes of acute myeloid leukemia. *Cancer Cell* **40**, 301317.e12 (2022).

Referees' report, third round of review

Reviewer #3: No additional comments.

Authors' response to the third round of review

Reviewer #3: No additional comments.

Thank you very much for your previous suggestions which we think have improved the paper.